# eQTL mapping in fetal-like pancreatic progenitor cells reveals early developmental insights into diabetes risk

Jennifer P. Nguyen[1,2], Timothy D. Arthur[2,3], Kyohei Fujita[4], Bianca M. Salgado[5], Margaret K. R. Donovan[1,2], iPSCORE Consortium*, Hiroko Matsui[5], Ji Hyun Kim [6], Agnieszka D'Antonio-Chronowska[4], Matteo D'Antonio[2,4,5] & Kelly A. Frazer [4,5] ✉

The impact of genetic regulatory variation active in early pancreatic development on adult pancreatic disease and traits is not well understood. Here, we generate a panel of 107 fetal-like iPSC-derived pancreatic progenitor cells (iPSC-PPCs) from whole genome-sequenced individuals and identify 4065 genes and 4016 isoforms whose expression and/or alternative splicing are affected by regulatory variation. We integrate eQTLs identified in adult islets and whole pancreas samples, which reveal 1805 eQTL associations that are unique to the fetal-like iPSC-PPCs and 1043 eQTLs that exhibit regulatory plasticity across the fetal-like and adult pancreas tissues. Colocalization with GWAS risk loci for pancreatic diseases and traits show that some putative causal regulatory variants are active only in the fetal-like iPSC-PPCs and likely influence disease by modulating expression of disease-associated genes in early development, while others with regulatory plasticity likely exert their effects in both the fetal and adult pancreas by modulating expression of different disease genes in the two developmental stages.

Genome-wide association studies (GWAS) have identified hundreds of genetic variants associated with adult pancreatic disease risk and phenotypes[1–4]. However, the majority of these associations map predominantly to non-coding regions of the genome, thereby hindering functional insights into disease processes[5–7]. Previous large-scale expression quantitative trait loci (eQTL) studies have made significant advancements toward understanding how genetic variation affects gene expression in various tissues and cell types, as well as their contribution to human traits and diseases[8–11]. However, these analyses were conducted in adult tissues and therefore the effects of regulatory variation on gene expression under fetal conditions remain unclear. Moreover, the integration of adult and fetal eQTL datasets would enable the investigation of regulatory plasticity of genetic variants,

which refers to changes in variant function under different spatio-temporal contexts[9,12,13]. Understanding how genetic variation affects gene expression during early pancreas development, and how their function changes in adulthood, can expand our understanding of the biological mechanisms underlying adult pancreatic disease and GWAS complex trait loci.

Many lines of evidence from clinical and genomic studies indicate an important role of pancreas development in the health and onset of childhood and adult pancreatic diseases[14–17]. For example, mutations in genes critical to pancreatic development, such as *PDX1*, *HNF4A*, and *HNF1A*, are associated with childhood-onset diabetes[18–20]. Furthermore, type 2 diabetes (T2D)-risk variants map to transcription factors (TFs) that are crucial to pancreas development, including NEUROG3

¹Bioinformatics and Systems Biology Graduate Program, University of California, San Diego, La Jolla, CA 92093, USA. ²Department of Biomedical Informatics, University of California, San Diego, La Jolla, CA 92093, USA. ³Biomedical Sciences Graduate Program, University of California, San Diego, La Jolla, CA 92093, USA. ⁴Department of Pediatrics, University of California, San Diego, La Jolla, CA 92093, USA. ⁵Institute of Genomic Medicine, University of California San Diego, 9500 Gilman Dr, La Jolla, CA 92093, USA. ⁶Department of Pediatrics, Dongguk University Ilsan Hospital, Goyang, South Korea. *A list of authors and their affiliations appears at the end of the paper. ✉e-mail: kafrazer@health.ucsd.edu

and HNF1A, and are enriched in accessible pancreatic progenitor-specific enhancers[4,14]. To address the limited availability of fetal pancreatic tissues, protocols have been developed to efficiently guide the differentiation of human induced pluripotent stem cells (iPSCs) into pancreatic progenitor cells (iPSC-PPCs)[21–28]. This approach serves as a model system to study human pancreas development. iPSC-PPCs demonstrate expression of key transcription factors associated with early pancreas development, including *PDX1*, *NKX6-1*, and *SOX9*, all pivotal for pancreas lineage specification and differentiation[22,29–33]. Additionally, iPSC-PPCs express developmental signaling pathway, including Notch, WNT, and Hedgehog, that are critical in pancreas development[34–38]. While iPSC-PPCs have provided extensive insights into pancreas developmental biology, they have not yet been utilized to examine the impact of genetic variation on gene expression in the fetal-like pancreas.

In this work, we conduct a large-scale eQTL analysis on 107 iPSC-PPC samples to map genetic loci associated with gene expression and isoform usage during early pancreas development. We integrate eQTLs from adult pancreatic tissues and identify eQTL loci that display temporal specificity in early pancreas development, as well as eQTL loci that are shared with adult but display regulatory plasticity. Annotation of GWAS risk loci using our spatiotemporally informed eQTL resource reveal candidate causal regulatory variants with developmental-unique effects associated with complex pancreatic traits and disease.

## Results

### Overview of study

The goal of our study is to understand how regulatory variation active during early pancreas development influences adult pancreatic disease risk and phenotypes (Fig. 1a). We differentiated 106 iPSC lines from the iPSCORE resource[39] derived from 106 whole-genome sequenced individuals to generate 107 iPSC-PPC samples (one iPSC line was differentiated twice) (Supplementary Fig. 1, Supplementary Data 1, Supplementary Data 2). We characterized the fetal-like pancreatic transcriptome as well as the cellular composition using single-cell RNA-seq (scRNA-seq) of eight iPSC-PPC samples. Then, we conducted an eQTL analysis on bulk RNA-seq of all 107 samples to identify regulatory variants associated with fetal-like pancreatic gene expression and isoform usage. To better understand the spatiotemporal context of genetic variants, we integrated eQTLs previously discovered in adult pancreatic islets[11] and whole pancreas[10] samples using colocalization and network analysis. Finally, using our eQTL resource of pancreas tissues (i.e., fetal-like iPSC-derived PPCs, adult islets, adult whole pancreas), we performed GWAS colocalization and fine-mapping to link developmental regulatory mechanisms and identify putative causal variants underlying pancreatic traits and disease associations.

### Large-scale differentiation of fetal-like pancreatic progenitor cells

We derived 107 iPSC-PPC samples using iPSC lines reprogrammed from 106 individuals. Differentiation efficiency was assessed using flow cytometry analysis on PDX1 and NKX6-1, which are two markers routinely assayed for early pancreatic progenitor formation. PDX1 marks the specification of cells towards the pancreas lineage (referred to here as "early PPC"; PDX1+/NKX6-1−), while subsequent NKX6-1 expression marks the differentiation and maturation of pancreatic progenitor cells (referred to here as "late PPC"; PDX1+/NKX6-1+)[40]. We observed an 18.1% median percentage of early PPCs (PDX1+/NKX6-1−) across the 107 samples while the median percentage of late PPCs (PDX1+/NKX6-1+) was 74% (range: 9.4–93.1%) (Fig. 1b, Supplementary Fig. 2, Supplementary Data 2). We further found that the median percentage of cells that expressed PDX1+ was more than 90%, confirming that the majority of cells have specified towards the pancreas lineage and that the differentiation procedure was highly efficient (Fig. 1b, Supplementary

Fig. 2, Supplementary Data 2). Consistent with flow cytometry analysis, scRNA-seq of ten derived iPSC-PPCs confirmed the presence of both early and late PPCs and that the majority of the cells were late PPCs (Supplementary Figs 3–8; Supplementary Data 2–5; See Methods and Supplemental Note 1). Altogether, these results show that the majority of the cells in iPSC-PPCs were differentiated into late PPCs while a smaller fraction represented a primitive PPC state.

To examine the similarities between iPSC-PPC and adult pancreatic transcriptomes, we generated bulk RNA-seq for all 107 iPSC-PPC samples and inferred the pseudotime on each sample, along with 213 iPSCs[39,41], 87 pancreatic islets[42], and 176 whole pancreatic tissues[43]. Pseudotime analysis and comparative expression analysis of early developmental genes showed that the iPSC-PPC samples corresponded to an early timepoint of pancreas development (Supplementary Fig. 9, Supplementary Data 6).

These analyses, combined with the results of previous studies[22,25,27,38], show that the 107 derived iPSC-PPCs represent a fetal-like state of pancreatic tissues.

### Identification and characterization of gene and isoform eQTLs in fetal-like iPSC-PPCs

To characterize the effects of genetic variation on the fetal-like iPSC-PPC transcriptome, we performed an eQTL analysis to map the genetic associations with fetal-like gene expression ($e_gQTL$) and relative isoform usage ($e_iQTL$). Considering only autosomal chromosomes, we analyzed a total of 16,464 genes and 29,871 isoforms (corresponding to 9624 autosomal genes) that were expressed in the fetal-like iPSC-PPCs. We identified 4065 (24.7%) eGenes and 4016 (13.0%) eIsoforms with an $e_gQTL$ or $e_iQTL$, respectively (FDR < 0.01, Fig. 1c, d, Supplementary Data 7). To identify additional independent eQTL signals (i.e., conditional eQTLs)[44], we performed a stepwise regression analysis for each eGene and eIsoform. This analysis yielded 368 $e_gQTLs$ that mapped to 338 eGenes and 216 $e_iQTLs$ that mapped to 198 eIsoforms, totaling to 4433 independent $e_gQTL$ associations and 4232 independent $e_iQTL$ associations (Fig. 1c, d, Supplementary Data 7). We next predicted candidate causal variants underlying each eQTL ($e_gQTL$ and $e_iQTL$) association using *coloc* genetic fine-mapping[45] and tested their enrichments in transcribed regions and regulatory elements. We observed an enrichment of $e_gQTLs$ in intergenic and promoter regions while $e_iQTLs$ were enriched in splice sites and RNA-binding protein binding sites (Fig. 1e). We additionally estimated the transcription factor (TF) binding score for each variant using the Genetic Variants Allelic TF Binding Database[46] and found that, at increasing posterior probability (PP, probability that the variant is causal for the association) thresholds, the candidate causal variants underlying $e_gQTLs$ were more likely to affect TF binding compared to those underlying $e_iQTLs$ (Fig. 1f, Supplementary Data 8). These results corroborate similar findings from previous studies[10,12,47], showing that the genetic variants underlying $e_gQTLs$ and $e_iQTLs$ primarily affect gene regulation and coding regions or alternative splicing, respectively.

To further characterize the function of genetic variants associated with the fetal-like iPSC-PPC transcriptome, we examined the distributions of $e_gQTLs$ and $e_iQTLs$ per gene. Of the 5619 genes whose phenotype was affected by genetic variation, 1008 were impacted through both gene expression and isoform usage (i.e., had both $e_gQTL$ and $e_iQTLs$, 17.9%) while 3057 were impacted through only gene expression (i.e., had only $e_gQTLs$, 54.4%) and 1554 through only isoform usage (i.e., had only $e_iQTLs$, 27.7%, Fig. 1g, Supplementary Data 7). For the 1008 genes with both $e_gQTL$ and $e_iQTLs$, we performed colocalization with *coloc.abf*[45] to examine whether the same or different genetic variants underpinned their associations. *Coloc.abf*[45] employs a Bayesian approach to estimate the PP that each of the five colocalization models best explains the association between two genetic signals: H0) no associations detected in either signal; H1) association detected in only signal 1; H2) association detected in only signal 2; H3) associations

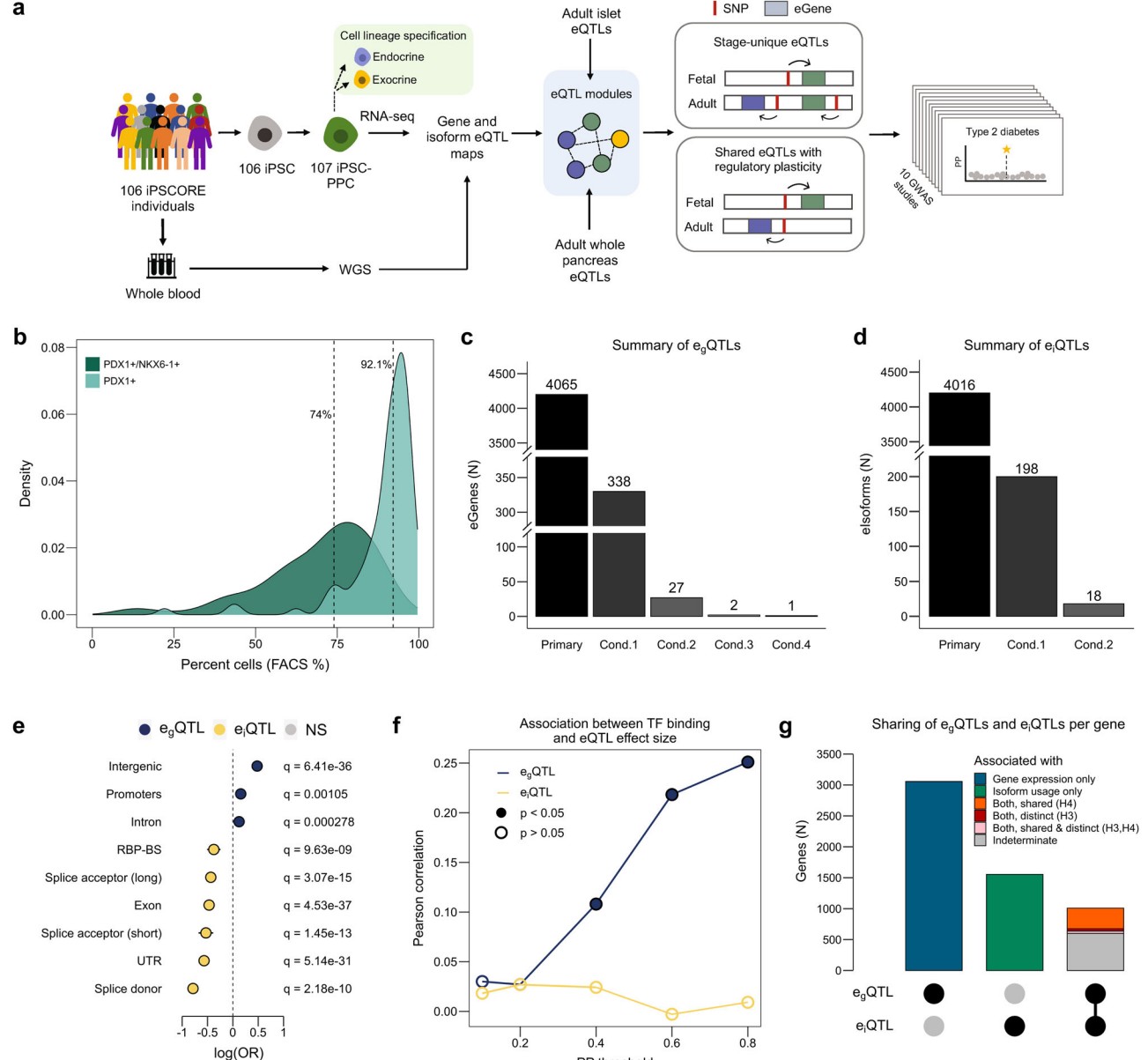

**Fig. 1 | Discovery and characterization of eQTLs in iPSC-PPC. a** Study overview created using PowerPoint. **b** Density plots showing the distribution of PDX1+ cells (%; regardless of NKX6-1 status; light green) and PDX1+/NKX6-1+ cells (%; dark green) in the 107 iPSC-PPC samples. **c** Bar plot showing the number of eGenes with primary and conditional $e_gQTLs$. **d** Bar plot showing the number of eIsoforms with primary and conditional $e_iQTLs$. **e** Enrichment (odds ratio, X-axis) of eQTLs in genomic regions (Y-axis) using two-sided Fisher's Exact Tests comparing the proportion of variants with causal posterior probability (PP) ≥ 5% in the genomic regions between $e_gQTLs$ (blue; $n = 8763$) and $e_iQTLs$ (yellow; $n = 8919$). **f** Line plot showing Pearson correlations of TF binding score and eQTL effect size at different thresholds of causal PP for $e_gQTLs$ (blue) and $e_iQTLs$ (yellow) (Supplementary Data 8). Closed

points indicate significant correlations (nominal $p < 0.05$) while open points indicate non-significant correlations (nominal ($p > 0.05$). **g** Bar plot showing the number of genes with only $e_gQTLs$ (blue; $n = 3057$), only $e_iQTLs$ (green; $n = 1554$), or both. Orange represents genes whose $e_gQTLs$ colocalized with all their corresponding $e_iQTLs$ (PP.H4 ≥ 80%; $n = 333$). Red represents genes whose $e_gQTLs$ did not colocalize with any of their corresponding $e_iQTLs$ (PP.H3 ≥ 80%; $n = 38$), and pink represents genes with both shared and distinct $e_gQTLs$ and $e_iQTLs$ (i.e., an eGene with two eIsoforms may colocalize with one eIsoform but not the other) ($n = 39$). Gray represents genes whose eQTL signals were not sufficiently powered to test for colocalization (PP.H4 < 80% and PP.H3 < 80%; $n = 598$).

detected in both signals but driven by different causal variants, and H4) associations detected in both signals and driven by the same causal variant. We identified 410 (40.7%) genes that had at least one H4 (PP.H4, posterior probability for H4 ≥ 80%) or H3 (PP.H3, posterior probability for H3 ≥ 80%) association between their $e_gQTL$ and $e_iQTLs$, of which the majority (333, 81.2%) had only overlapping signals (all H4), 38 (9.3%) had only non-overlapping signals (all H3), and 39 (9.5%) had both overlapping and non-overlapping $e_iQTLs$ (both H3 and H4; an $e_gQTL$ can colocalize with an $e_iQTL$ corresponding to one isoform but not with another $e_iQTL$ corresponding to a second isoform) (Fig. 1g,

Supplementary Data 9). The remaining 598 genes had PP.H3 < 80% and PP.H4 < 80% due to insufficient power (Fig. 1g). These findings show that 19.5% (1008/5169) of genes had both $e_gQTLs$ and $e_iQTLs$ and that their effects were commonly driven by the same causal variants (81.2%) while only a small fraction was driven by different causal variants (9.3%). Enrichment analysis of overlapping $e_gQTL$ and $e_iQTLs$ showed that these variants were enriched for stop codons as well as mechanisms affecting both gene expression and alternative splicing (Supplementary Fig. 11). Overall, our results show that the majority of genes had either only $e_gQTLs$ or $e_iQTLs$, indicating that the functional

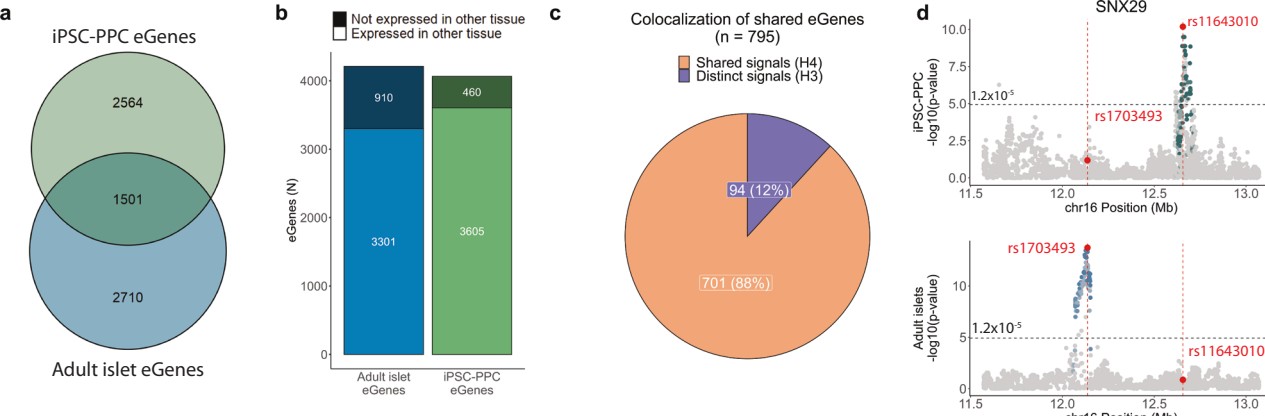

**Fig. 2 | Comparison of the genetic architecture underlying gene expression between fetal-like and adult islets. a** Venn diagram showing the overlap of eGenes between fetal-like iPSC-PPC and adult islets. **b** Stacked bar plot showing the total number of eGenes detected in adult islets (blue; *n* = 4211 total) that were expressed in iPSC-PPC (light blue; *n* = 3301). Likewise, we show the total number of fetal-like iPSC-PPC eGenes (green; *n* = 4065 total) that were expressed in adult islets (light green; *n* = 3605). These results show that the majority of eGenes were expressed in both tissues, however, a large fraction was influenced by genetic variation in only one of the two tissues. Therefore, the small overlap of eGenes may be due to differences in the genetic regulatory landscape. **c** Pie chart showing the proportion of shared eGenes with distinct genetic loci (PP.H3 ≥ 80%, purple) or shared genetic

loci (PP.H4 ≥ 80%, orange). These results show that 12% of the shared eGenes were associated with distinct regulatory variants between fetal-like and adult pancreatic stages. **d** Example of a shared eGene (*SNX29*) whose expression was associated with distinct $e_gQTL$ signals (PP.H3 = 90.4%) in fetal-like iPSC-PPC (green, top panel) and adult islets (blue, bottom panel). The X-axis represents variant positions while the Y-axis shows the −log10(eQTL *p*-value) for the associations between the genotype of the tested variants and gene expression. For plotting purposes, we assign a single *p*-value for gene-level significance after Bonferroni-correction (0.05/number of independent variants tested in fetal-like iPSC-PPC; horizontal line). Red vertical lines show the positions of the lead variants in fetal-like iPSC-PPC and adult islets (chr16:12656135:C > G and chr16:12136526:A > G, respectively).

mechanisms underlying these associations are likely independent, where genetic variants affecting alternative splicing do not affect the overall expression of the gene, and vice versa.

## eQTL landscapes of fetal-like iPSC-PPC and adult pancreatic islets

Studies aimed at identifying and characterizing eGenes have been conducted in both adult human islets and whole pancreatic tissues;[8,10,11,42,48] however, islet tissues have been more thoroughly studied because of their role in diabetes. Therefore, we focused on understanding the similarities and differences between eGenes in the fetal-like iPSC-PPCs and adult human islets.

We obtained eQTL summary statistics and intersected the 4211 autosomal eGenes identified in 420 adult islet samples[11] with the 4065 eGenes in fetal-like iPSC-PPC. We found that only 1501 (36.9% of 4065) eGenes overlapped between the fetal-like iPSC-PPC and adult islet tissues (Fig. 2a). To determine whether the small overlap was due to gene expression differences, we calculated how many of the eGenes were expressed in both the fetal-like iPSC-PPC and adult islets. Of the 4065 fetal-like iPSC-PPC eGenes, 88.7% (3605) were also expressed in the adult islets; likewise, of the 4211 adult islet eGenes, 78.4% (3301) were also expressed in the fetal-like iPSC-PPCs (Fig. 2b). These results suggest that most fetal-like iPSC-PPC eGenes were expressed but not associated with genetic variation in the adult islet samples, and vice versa.

For eGenes that were present in both the fetal-like iPSC-PPC and adult islet samples, we next asked whether their expressions were controlled by the same genetic variants. We performed colocalization between $e_gQTLs$ for the 1501 shared eGenes in the fetal-like iPSC-PPC and adult islets and found that 795 (52.3%) displayed strong evidence for either H3 or H4 association (PP.H3 or PP.H4 ≥ 80%) (Supplementary Data 9). Of the 795 with an association, 701 (88.2%) had overlapping $e_gQTL$ signals (PP.H4 ≥ 80%) while 94 (11.8%) had non-overlapping $e_gQTL$ signals (PP.H3 ≥ 80%) (Fig. 2c, Supplementary Data 9). These results indicate that most shared eGenes were associated with the same genetic variants controlling their gene expressions in both fetal-like iPSC-PPC and adult islet tissues, while a subset

had non-overlapping genetic variants. For example, we identified *SNX29* as an eGene in both fetal-like iPSC-PPC and adult islets but observed that its expression was associated with distinct eQTL signals approximately 520 kb apart (Fig. 2d). SNX29 is involved in various signaling pathways[49], including TGF-*β*, ErbB, and WNT signaling pathways, and is predicted to be a causal gene for body-mass index (BMI) and T2D[50–52]

Taken together, our results show that a minor proportion of fetal-like iPSC-PPC eGenes (1501, 37% of 4065) was shared with adult islets, whereas the majority (2564 = 4065−1501, 63%) were fetal development-specific; and, while most shared eGenes were associated with the same regulatory variants, -12% were mediated by different eQTLs. These findings indicate that regulatory variants tend to act in a developmental-specific manner, potentially by affecting the binding of key regulatory TFs specific to fetal or adult pancreatic stages.

## Developmental stage-unique and shared $e_gQTLs$

Above, we described eGenes that were unique to either fetal-like iPSC-PPCs or adult islets, or shared between both. Here, we sought to identify eQTLs (i.e., regulatory variants) that specifically affect gene expression during the pancreas development stage, in the adult stage, or both stages. Because fetal-like iPSC-PPCs give rise to both endocrine and exocrine cell fates, we included eQTLs from both adult islets[11] and whole pancreas[10] tissues in our analyses. Due to the many different types of eQTLs used in this study, we refer to all eQTLs as a collective unit as "eQTLs", eQTLs that were associated with gene expression as "$e_gQTLs$" (as defined above), and eQTLs associated with changes in alternative splicing (eᵢQTLs, exon eQTLs, and sQTLs) as "$e_{AS}QTLs$". For simple interpretations, we only describe the results for the analyses conducted on the $e_gQTLs$ below, however, we identified unique and shared iPSC-PPC $e_{AS}QTL$ associations by conducting the same analyses (see Supplementary Note 2).

To identify $e_gQTLs$ that shared the same regulatory variants, we performed pairwise colocalization using *coloc.abf*[45] between $e_gQTLs$ in fetal-like iPSC-PPC, adult islets[11], and adult whole pancreas samples[10]. We considered only $e_gQTLs$ that had at least one variant with causal PP ≥ 1% (from genetic fine-mapping[45]), were outside the

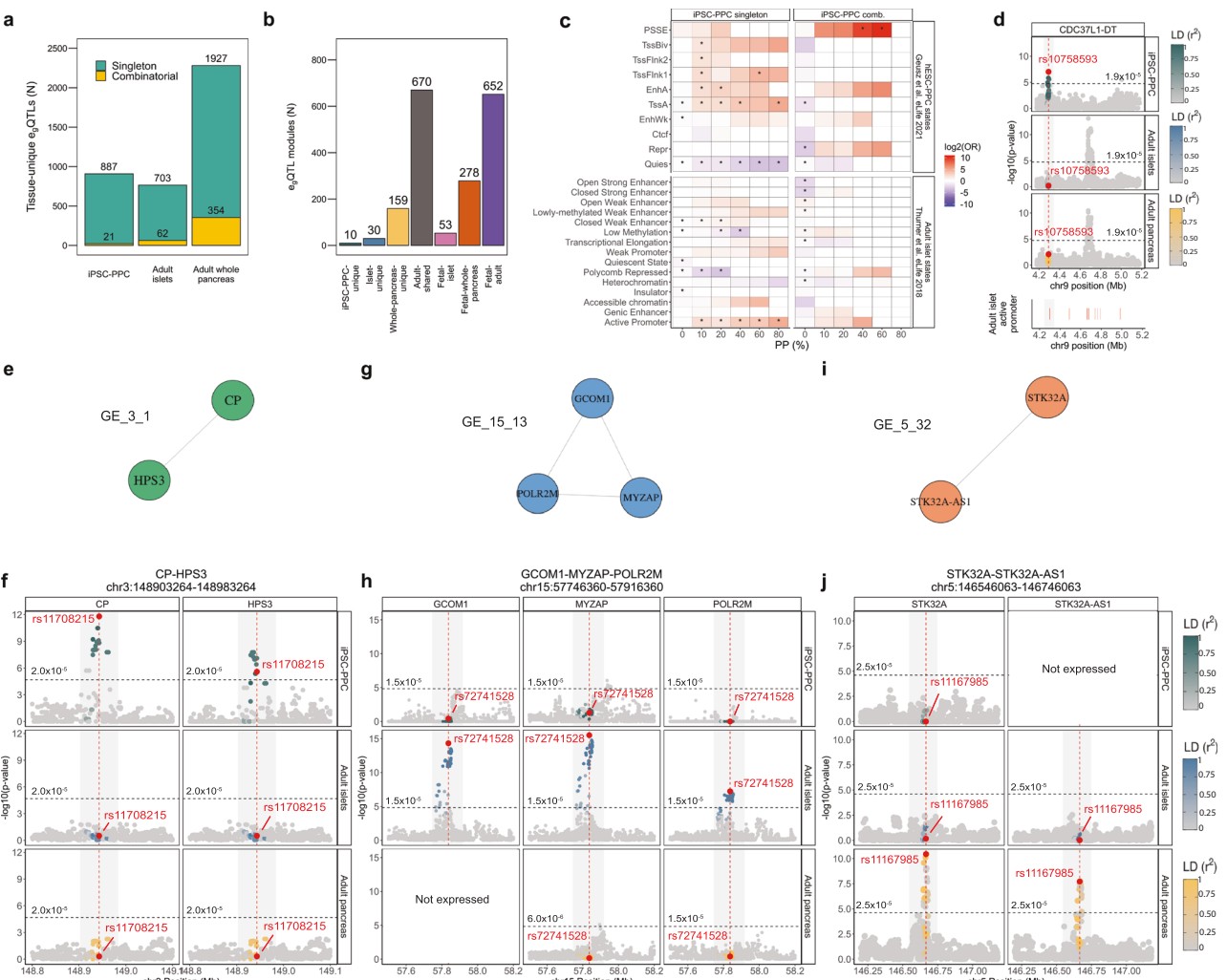

**Fig. 3 | eQTL sharing between iPSC-PPC, adult islets, and adult whole pancreas. a** Bar plot showing the number of tissue-unique e$_g$QTLs identified in fetal-like iPSC-PPC, adult islets and adult whole pancreas. **b** Bar plot showing the number of e$_g$QTL modules for each annotation. **c** Top panels: Enrichment (odds ratio) of iPSC-PPC singleton and combinatorial e$_g$QTLs in hESC-derived PPC chromatin states[14]. Bottom panels: Enrichment (odds ratio) of iPSC-PPC singleton and combinatorial e$_g$QTLs in adult islet chromatin states[54]. Enrichment was calculated using a two-sided Fisher's Exact Test comparing the proportion of candidate causal variants overlapping the chromatin states versus a background of randomly selected 20,000 variants at various PP thresholds. *P*-values were Benjamini-Hochberg-corrected and considered significant if the corrected *p*-values < 0.05 (indicated by asterisk, Supplementary Data 12). **d** *CDC37L1-DT* locus showing an iPSC-PPC-unique singleton e$_g$QTL overlapping an adult islet active promoter region. Lower panel shows the positions of active promoters in the adult islets. **e, f** The chr3:148903264-148983264 locus (gray rectangle) is an example of an "iPSC-PPC-unique" module (module ID: GE_3_1) associated with *CP* and *HPS3* expression. **g, h** The

chr15:57746360-57916360 locus (gray rectangle) is as an example of an "adult islet-unique" module (module ID: GE_15_13) associated with *GCOM1*, *MYZAP*, and *POLR2M* expression. *GCOM1* was not expressed in adult whole pancreas and therefore, was not tested for e$_g$QTL association. **i, j** The chr5:146546063-146746063 locus (gray rectangle) is an "adult whole pancreas-unique" e$_g$QTL module (module ID: GE_5_32) associated with *STK32A* and *STK32A-AS1* expression. *STK32A-AS1* was not expressed in iPSC-PPC and therefore, was not tested for e$_g$QTL association. Panels **e, g, i** display the e$_g$QTL modules as networks in which the e$_g$QTL associations (nodes) are connected by edges due to colocalization (PP.H4 ≥ 80%). For panels **d, f, h**, and **j**, the X-axis represents variant positions while the Y-axis shows the −log10(eQTL *p*-value) for the associations between the genotype of the tested variants and gene expression. For plotting purposes, we assigned a single *p*-value for gene-level significance after Bonferroni-correction (0.05/the number of independent variants tested in fetal-like iPSC-PPC; horizontal line). Red vertical lines indicate the positions of the lead candidate causal variants underlying the colocalization based on maximum PP.

MHC region, and associated with genes annotated in GENCODE version 34[53] (Supplementary Data 10). From colocalization, we identified 7893 pairs of e$_g$QTLs that displayed high evidence of colocalization with PP.H4 ≥ 80%, and 8570 e$_g$QTLs that did not colocalize with e$_g$QTLs (Supplementary Data 9). Hereafter, we refer to eQTLs that did not colocalize with eQTLs as "singletons" and those that colocalized with another eQTL (PP.H4 ≥ 80%; same or different tissue) as "combinatorial" (i.e., the 7893 pairs of e$_g$QTLs).

We next sought to identify singleton and combinatorial e$_g$QTL signals that were unique to iPSC-PPC or shared between iPSC-PPC and the adult pancreatic tissues. The singleton iPSC-PPC e$_g$QTLs were associated with a single eGene and not active in the adult pancreatic

samples, and hence tissue-unique. To ensure that there was no overlap of singleton e$_g$QTLs with other e$_g$QTLs in either the fetal-like and adult pancreas tissues, we implemented an LD filter (r² ≥ 0.2 with any e$_g$QTL within 500 Kb or within 500 Kb if LD could not be calculated; see Methods). We identified 3517 tissue-unique singleton e$_g$QTLs (887 iPSC-PPC + 703 adult islet + 1927 adult whole pancreas) that were not in LD with any nearby e$_g$QTLs (Fig. 3a, Supplementary Data 10).

To identify tissue-unique combinatorial e$_g$QTL signals, we created a network using the 7893 pairs of colocalized e$_g$QTL associations. We identified 1852 e$_g$QTL modules that passed specific criteria for module identification and LD filters (see Methods), of which 939 (50.7%) comprised two e$_g$QTLs while the remaining 913 (49.3%) had an average of

four $e_g$QTLs per module (range: 3-20 $e_g$QTLs) (Supplementary Data 10, Supplementary Data 11). In total, we identified 199 (10.7% of 1852) modules that were tissue-unique, of which 10 were iPSC-PPC-unique, 30 adult islet-unique, and 159 adult whole pancreas-unique (Fig. 3b), and altogether comprised 21, 62, and 354 $e_g$QTLs in combinatorial associations, respectively (Fig. 3a). In contrast, the remaining 1653 (89.3% of 1852) modules were associated with multiple pancreatic tissues, of which 670 were shared between only adult islet and whole pancreas tissues (referred to as "adult-shared"), 53 were shared between only iPSC-PPC and adult islets ("fetal-islet"), 278 between only iPSC-PPC and adult whole pancreas ("fetal-whole-pancreas"), and 652 between all three pancreatic tissues ("fetal-adult") (Fig. 3b). Together, the 983 (53 + 278 + 652) modules shared between iPSC-PPC and an adult pancreatic tissue were composed of 1122 iPSC-PPC, 870 adult islets, and 1394 adult whole pancreas $e_g$QTLs (Supplementary Data 10, Supplementary Data 11).

For $e_{AS}$QTLs, we observed similar trends in which the majority of fetal-like iPSC-PPC-unique $e_{AS}$QTLs were singletons and that combinatorial $e_{AS}$QTLs were likely shared with the adult pancreas tissues (see Supplemental Note 2; Supplementary Fig. 11). Altogether, including $e_{AS}$QTLs, we identified 1805 iPSC-PPC eQTLs that were unique to fetal-like iPSC-PPC, of which 1518 (887 $e_g$QTLs + 631 $e_{AS}$QTLs) functioned as singletons and 287 (21 $e_g$QTLs + 266 $e_{AS}$QTLs) in modules; while 1977 (1175 $e_g$QTLs + 802 $e_{AS}$QTLs) were shared with adult pancreatic tissues, and 4326 (2066 $e_g$QTLs + 2260 $e_{AS}$QTLs) failed one or more the stringent filters and were marked as ambiguous (Supplementary Data 10, Supplementary Data 11).

## Functional enrichment and characterization of fetal-like iPSC-PPC-unique $e_g$QTLs

To functionally characterize the fetal-like iPSC-PPC tissue-unique singleton and combinatorial $e_g$QTLs, we calculated their enrichments in chromatin state annotations from human ESC-derived PPCs[14]. At high PP thresholds, we observed the strongest enrichment of singleton $e_g$QTLs in active promoter (TssA) regions, consistent with their role in regulating the expression of a single gene (Fig. 3c, Supplementary Data 12). For combinatorial $e_g$QTLs, we observed a strong enrichment in PPC-specific stretch enhancer (PSSE) regions at high PP thresholds ($p = 0.001$, OR = 1345, PP threshold = 60%) (Fig. 3c, Supplementary Data 12), consistent with their involvement in the transcriptional regulation of multiple genes. We also evaluated the enrichment of fetal-like iPSC-PPC-specific singleton and combinatorial $e_g$QTLs in adult islet chromatin states[54] (Fig. 3c). No meaningful enrichments were observed for fetal-like iPSC-PPC-unique combinatorial $e_g$QTLs, but iPSC-PPC-unique singleton $e_g$QTLs were enriched in adult promoter regions ($p = 4.1 \times 10^{-4}$, OR = 28.4, PP threshold = 80%). For example, we observed that the iPSC-PPC-unique singleton $e_g$QTL in the *CDC37L1-DT* locus overlapped an active promoter region in the adult islet, while in both adult islet and adult whole pancreas, the variants in the same region are not active (Fig. 3d). Overall, these results show that the $e_g$QTLs annotated as iPSC-PPC tissue-unique were enriched in regulatory elements consistent with their proposed functions.

Next, we present three examples of tissue-unique $e_g$QTL modules that further illustrate context-specificity of regulatory variants in the three pancreatic tissues. We identified the $e_g$QTL module GE_3_1 ("GE" means that this module is associated with gene expression) as a fetal-unique $e_g$QTL locus (ch3:148903264-148983264) because the underlying genetic variants were associated with *CP* and *HPS3* expression in only fetal-like iPSC-PPC while in adult islets and whole pancreas, the variants were not detected as $e_g$QTLs (Fig. 3e, f). Similarly, GE_15_13 was an adult islets-unique $e_g$QTL locus (chr15:57746360-57916360) associated with *GCOM1, MYZAP*, and *POLR2M* expression, while in the other two pancreatic tissues, the variants were inactive and not associated with gene expression (Fig. 3g, h). Finally, we discovered GE_5_32 as an adult whole pancreas-unique $e_g$QTL locus (chr5:146546063-

146746063) associated with *STK32A* and *STK32A-AS1* expression in only the adult whole pancreas (Fig. 3i, j). Together, these results show that gene regulation varies between fetal-like and adult pancreatic stages, as well as between the two adult tissues, further demonstrating the importance of profiling multiple contexts of the pancreas to delineate molecular mechanisms underlying pancreatic disease.

## Regulatory plasticity in combinatorial $e_g$QTLs shared between fetal-like and adult pancreatic tissues

Regulatory elements are known to have context-specific gene interactions[55]. To explore this further, we examined the 983 $e_g$QTL modules shared between fetal-like iPSC-PPC and adult pancreatic tissues and determined whether the modules were associated with the same or different eGenes between the two stages. We characterized the eGene overlap in five different ways (Fig. 4a, Supplementary Data 11): A) 200 (20.3%) $e_g$QTL modules were associated with same eGene(s) (range: 1-2) between fetal-like iPSC-PPC and only one of the two adult pancreatic tissues; B) 305 (31.0%) were associated with the expression of the same eGene(s) (range 1-2) in the fetal-like and both adult tissues; C) 350 (35.6%) were associated with 2-12 eGenes, some of which were shared, but at least one eGene was different between the fetal-like and at least one of the adult tissues (referred to as "partial overlap"); D) 88 (9.0%) were associated with different eGenes (range: 2-5) between fetal-like iPSC-PPCs and one of the two adult pancreatic tissues; and E) the remaining 40 (4.1%) were associated with different eGenes (range: 2-7) between the fetal-like and both adult islet and whole pancreas tissues (i.e., there is no overlap of eGenes between the two developmental stages). These data show that 51.3% (505, categories A and B) of the modules shared between the iPSC-PPCs and adult pancreatic tissues regulated expression of the same genes, while 48.7% (478, categories C-E) displayed spatiotemporal regulatory plasticity.

Here, we illustrate examples of $e_g$QTL modules in three intervals that display regulatory plasticity between fetal-like and adult states. In the chr1:201300813-201450813 locus, we identified a fetal-adult $e_g$QTL module (GE_1_163) that comprised $e_g$QTL associations for different eGenes in iPSC-PPC and the two adult pancreatic tissues, specifically *AC119427.1* in iPSC-PPC and *TNNI1* in the two adult tissues (Fig. 4b). Likewise, the chr19:4213666-4433666 locus corresponding to a fetal-adult $e_g$QTL module (GE_19_90) was associated with *MPND* expression in only iPSC-PPC but in adult islets and whole pancreas, the underlying variants were associated with *STAP2* expression (Fig. 4c). Finally, the fetal-adult $e_g$QTL locus (GE_10_11) in chr10:1273918-1276118 affected *UROS* expression in all three pancreatic tissues but in adult islets, the underlying variants also affected *BCCIP* expression (Fig. 4d). Together, these genomic loci illustrate examples of regulatory plasticity observed in genetic variants in which their genotypes incur different impacts on transcriptional activity depending on the life stage of the pancreas.

Altogether, including $e_{AS}$QTLs, we discovered 655 (478 $e_g$QTL + 177 $e_{AS}$QTL modules, categories C-E) shared eQTL loci that displayed regulatory plasticity in which the underlying regulatory variants were associated with one or more different genes and could thereby affect different biological processes (see Supplemental Note 2, Supplementary Fig. 11, Supplementary Data 11). These 655 shared eQTL loci comprise 1043 iPSC-PPC, 934 adult islet, and 1111 adult whole pancreas eQTL associations (Supplementary Data 10).

## Associations of spatiotemporal eQTLs with pancreatic traits and disease phenotypes

To better understand the role of regulatory variants associated with complex human traits and disease during early development and adult pancreatic stages, we performed colocalization between GWAS signals and eQTLs ($e_g$QTL and $e_{AS}$QTL) detected in

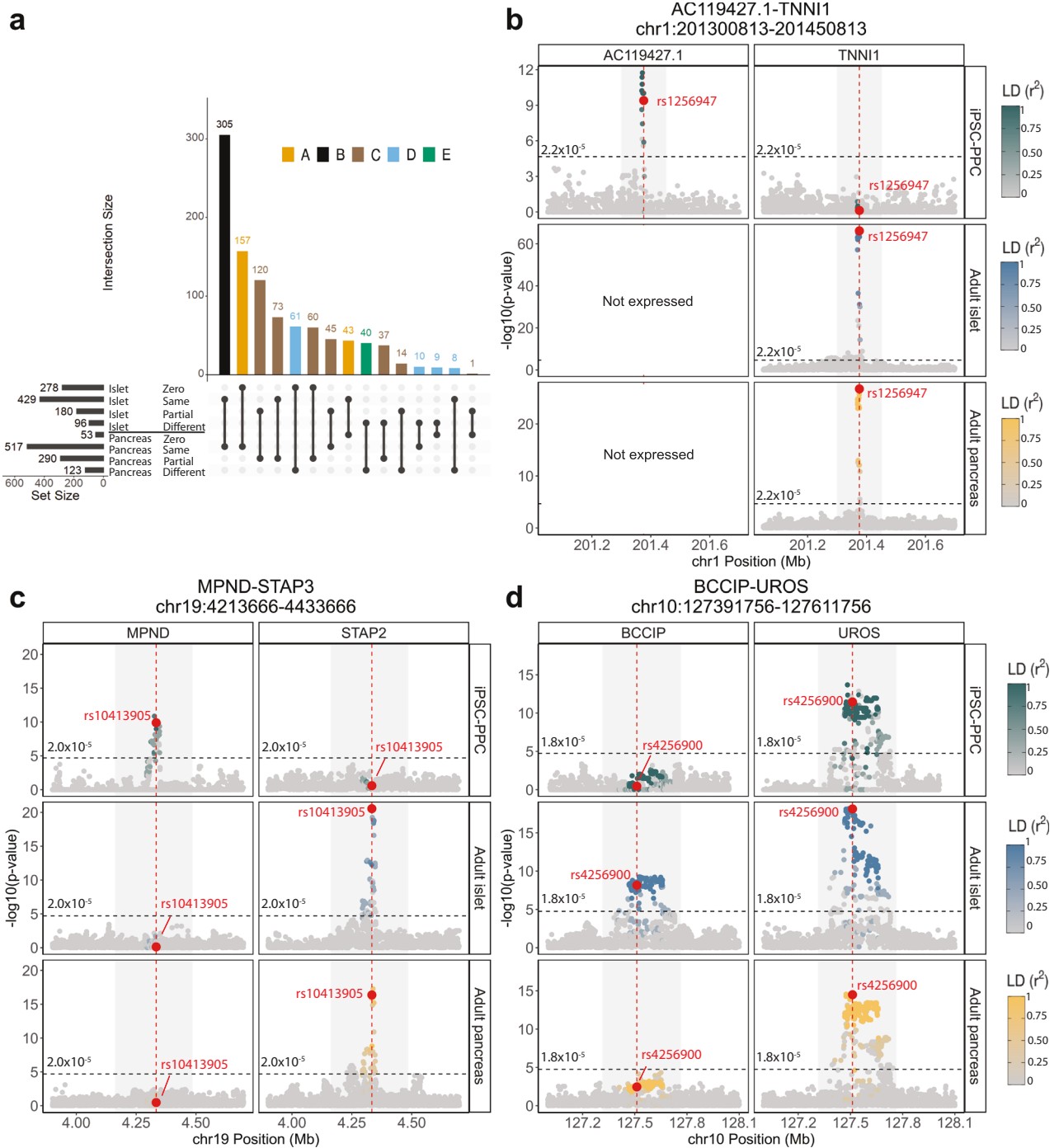

**Fig. 4 | Regulatory plasticity of e$_g$QTL loci. a** Number of e$_g$QTL modules shared between iPSC-PPC and at least one adult pancreas tissue categorized by eGene overlap with adult. "Zero" indicates that the module did not contain an e$_g$QTL in the respective adult tissue. "Same" indicates that the module had e$_g$QTLs for only the same eGenes in iPSC-PPC and the adult tissue. "Partial" indicates that the module had e$_g$QTLs for partially overlapping eGenes between iPSC-PPC and the adult tissue. "Different" indicates that the module had e$_g$QTLs for only different eGenes between iPSC-PPC and the adult tissue. **b–d** Examples of e$_g$QTL loci demonstrating regulatory plasticity of genetic variation across fetal-like and adult pancreatic stages. Panel **b** shows a locus strongly associated with *AC119427.1* expression in fetal-like iPSC-PPC and *TNNI1* expression in adult islet and whole pancreas. Panel **c** shows a

locus associated with *MPND* expression in only fetal-like iPSC-PPC but *STAP2* expression in both the adult pancreatic tissues. Panel **d** shows a locus associated with partially overlapping eGenes between the two pancreatic stages (*UROS* in all three pancreatic tissues and *BCCIP* in only adult islets). The X-axis represents variant positions while the Y-axis shows the −log10(eQTL *p*-value) for the associations between the genotype of the tested variants and gene expression. For plotting purposes, we assigned a single *p*-value for gene-level significance after Bonferroni-correction (0.05/the number of independent variants tested in fetal-like iPSC-PPC; horizontal line). Red vertical lines indicate the positions of the lead candidate causal variants underlying the colocalization based on maximum PP.

fetal-like iPSC-PPC, adult islets, and adult whole pancreas tissues. For this analysis, we considered GWAS data from ten different studies for two diseases involving the pancreas, including type 1 diabetes (T1D)[3] and type 2 diabetes (T2D)[4], and seven biomarkers

related to three traits: 1) glycemic control (HbA1c levels and fasting glucose [FG])[2,56]; 2) obesity (triglycerides, cholesterol, HDL level, and LDL direct)[56]; and 3) body mass index (BMI)[56] (Supplementary Data 13).

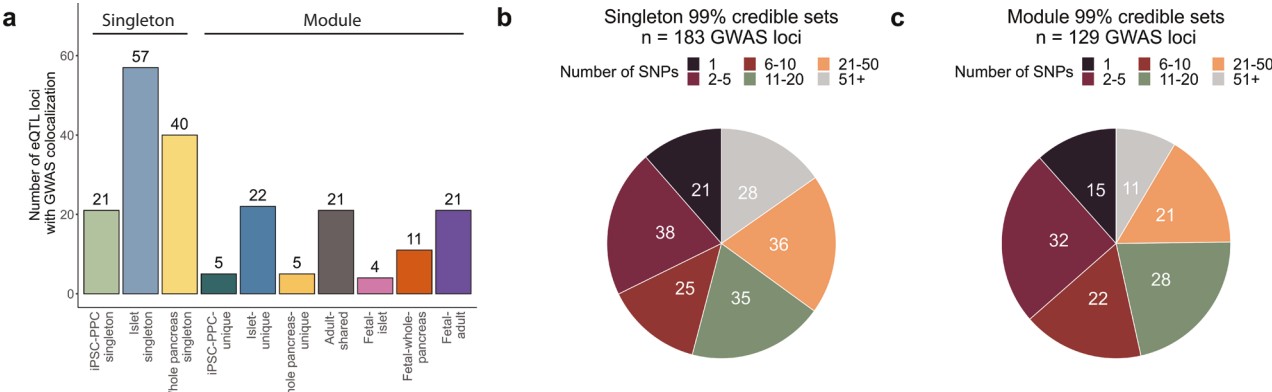

**Fig. 5 | Summary of pancreatic GWAS associations. a** Bar plot showing the number of eQTL loci that colocalized with GWAS variants (PP.H4 ≥ 80%) as a singleton or module. **b** Pie chart showing the number of singleton-colocalized GWAS loci (*n* = 183) color-coded by the number of candidate causal variants identified in their 99% credible sets. **c** Pie chart showing the number of module-colocalized GWAS loci (*n* = 129) color-coded by the number of candidate causal variants identified in their 99% credible sets.

**Singleton eQTLs**. Out of the 6101 singleton eQTLs (3517 e_gQTLs and 2584 e_ASQTLs; see Supplemental Note 2) in the fetal-like iPSC-PPC and two adult pancreatic tissues, we found 118 (1.9%) that displayed strong evidence for colocalization with at least one GWAS signal, including 21 (of 1518 total singleton eQTLs; 1.4%) fetal-like iPSC-PPC, 57 (of 2225; 2.6%) adult islets, and 40 (of 2358; 1.7%) adult whole pancreas singleton eQTLs (Fig. 5a, Supplementary Data 13). Given that some traits were highly correlated with one another[57,58], we observed 38 singleton eQTLs that colocalized with GWAS variants associated with more than one trait (range: 2-6 traits). In total, we identified 183 GWAS loci across the ten traits that colocalized with fetal-like or adult pancreatic singleton eQTLs (each combination of colocalized eQTL-GWAS trait variants was counted as a separate locus; Supplementary Fig. 12, Supplementary Fig. 13, Supplementary Data 13). We next identified putative causal variants underlying both eQTL and trait associations using *coloc.abf* [45] and constructed 99% credible sets (i.e., set of variants with a cumulative causal PP ≥ 99%; see Methods). Of the total 183 colocalized GWAS loci, we resolved 21 to a single putative causal variant while 63 had between two and ten variants and the remaining 99 had more than ten variants with an average of ~46 variants per locus (Fig. 5b, Supplementary Data 14).

**eQTL modules**. We next analyzed the combinatorial eQTLs for GWAS colocalization. We considered an eQTL module to overlap with GWAS variants if more than 30% of the eQTLs in the module colocalized with PP.H4 ≥ 80% and the number of H4 associations was twice greater than the number of H3 associations (see Methods). Of the 2832 (1852 e_gQTL and 980 e_ASQTL) modules, 89 (57 e_gQTL + 32 e_ASQTL; 3.1%) colocalized with a total of 129 GWAS loci across the ten traits (Supplementary Fig. 12, Supplementary Fig. 13, Supplementary Data 13). Of these 89 GWAS-colocalized modules, 5 were iPSC-PPC-unique, 36 were shared between both iPSC-PPC and adult, (4 fetal-islet, 11 fetal-whole-pancreas, and 21 fetal-adult modules), and 48 were associated with only adult (22 islet-unique, 5 whole pancreas-unique, 21 adult-shared) (Fig. 5a, Supplementary Fig. 13). We observed that all 5 iPSC-PPC-unique eQTL modules corresponded to e_ASQTL modules. This finding aligns with multiple studies showing that alternative splicing is more dynamic and extensive in embryonic and fetal stages[59–61]. The 89 modules comprised 49 iPSC-PPC eQTLs (41 genes), 98 adult islets eQTLs (75 genes), and 71 adult whole pancreas eQTLs (69 genes). To fine-map each of the 129 colocalized GWAS loci, we used the eQTL in the module that resulted in the least number of putative causal variants (see Methods). 15 GWAS loci had a credible set size of one variant, 54 with two to ten variants, and the

remaining 60 had more than ten variants and an average of ~32 variants per set (Fig. 5c, Supplementary Data 14).

Altogether, these results show complex pancreatic disease and trait GWAS variants colocalized with regulatory variants that were uniquely active in either the fetal-like or adult developmental stages and with regulatory variants shared across the life stage of the pancreas. Furthermore, our data show the utility of using spatiotemporally informed eQTLs for fine-mapping causal variants in GWAS loci.

**Spatiotemporally informed eQTL resource provides mechanistic insights into GWAS signals**
To assess the utility of our spatiotemporally informed eQTL resource for interpreting GWAS signals, we initially examined the role of regulatory plasticity in pancreatic disease and traits. We examined the 36 eQTL modules that were shared between fetal-like iPSC-PPC and the adult pancreatic tissues (i.e., fetal-adult, fetal-islet, and fetal-whole-pancreas) and colocalized with GWAS signals. Thirty of these modules were associated with the same genes (categories A and B), while one was associated with partially overlapping eGenes (category C), and five were associated with entirely different eGenes (category D and E) (see Fig. 4a for category definitions) (Supplementary Data 13). These results show that while the function of shared GWAS regulatory variants tended to be conserved across the fetal-like and adult pancreatic stages, a subset (17%, *n* = 6 of 36 total eQTL modules) were associated with distinct genes between the two stages.

To further assess the utility of our spatiotemporally informed eQTL resource, we next examined GWAS signals that could only be interpreted by including fetal pancreatic eQTLs. We calculated the fraction of GWAS loci that colocalized with only iPSC-PPC eQTLs, only adult islet eQTLs, and only adult whole pancreas eQTLs. For fair comparisons, we considered only e_gQTLs in this assessment. Of the 191 GWAS loci that colocalized with an e_gQTL (Supplementary Data 13), we found that 13% (24 loci) colocalized with 16 iPSC-PPC-unique e_gQTLs, 25% (47) with 27 adult islet-unique e_gQTLs, and 28% (53) with 46 adult whole pancreas-unique e_gQTLs. The remaining 35% (67) GWAS loci colocalized with 121 e_gQTLs shared between multiple tissues. We next determined how many of the 16 iPSC-PPC-unique e_gQTLs were active in 48 non-pancreatic tissues in the GTEx study[10]. We calculated LD ($r^2 > 0.2$ within 500 Kb or within 500 Kb if LD information was not available) between the lead variants of each of the 16 iPSC-PPC-unique e_gQTL and each e_gQTL for the non-pancreatic tissues in GTEx. We identified 8 (31% of 16, all singletons) that were independent and exclusive to the fetal-like iPSC-PPC dataset (i.e., did not have LD; Supplementary Data 15). One of these 8 iPSC-PPC e_gQTLs (*TPD52* e_gQTL) is described below in further detail. These results show that

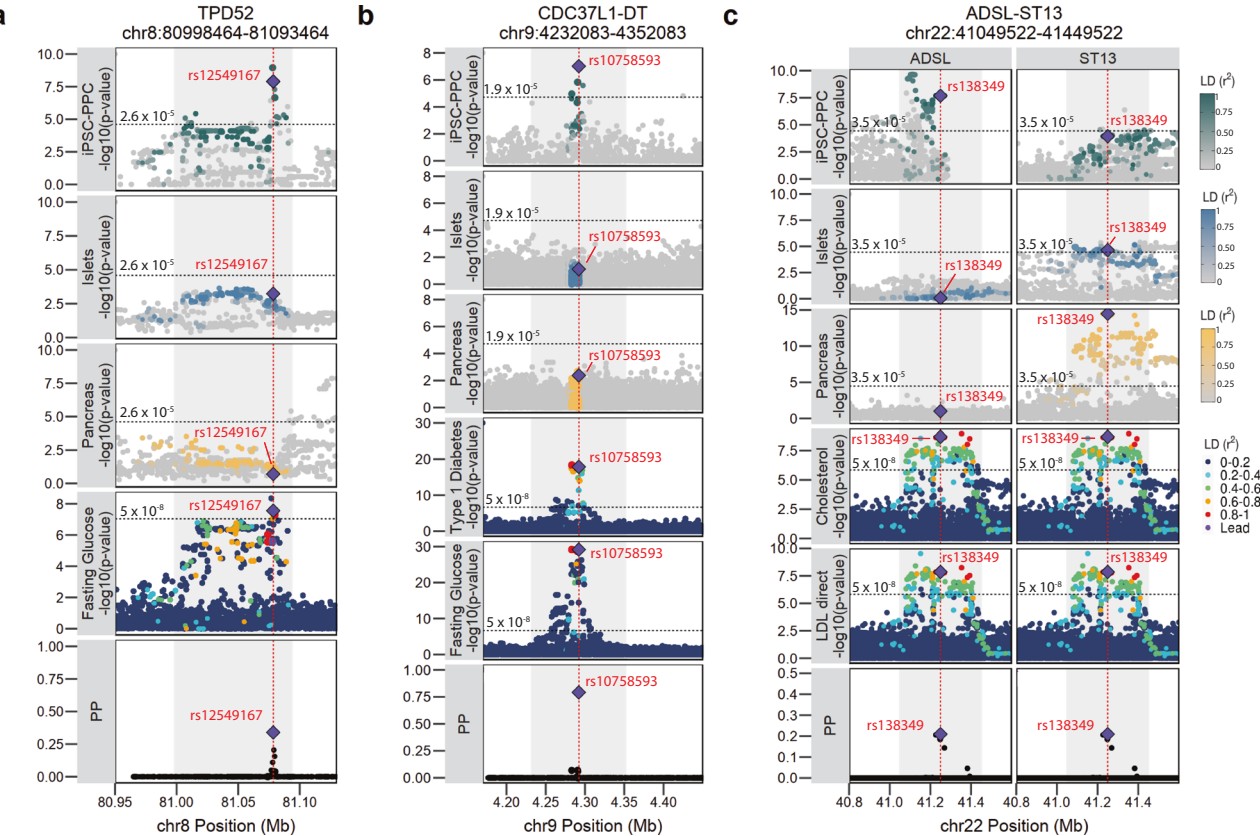

**Fig. 6 | Pancreatic GWAS associations with fetal-specific and adult-shared gene Expression. a** The TPD52 locus is associated with fasting glucose levels and colocalized with a fetal-like iPSC-PPC-unique singleton $e_g$QTL with the predicted causal variant identified as rs12549167 (chr8:81078464:C > T, PP = 33.9%). **b** The CDC37L1-DT locus is associated with fasting glucose and type 1 diabetes and colocalized with an iPSC-PPC-unique singleton $e_g$QTL with the predicted causal variant identified as rs10758593 (chr9:4292083:G > A, PP = 79.2%). **c** Cholesterol and LDL direct GWAS loci colocalize with a fetal-adult $e_g$QTL module where the variants are strongly associated with *ADSL* expression in iPSC-PPC and *ST13* expression in the adult whole pancreas (also weakly associated with *ST13* expression in the adult islets). The predicted causal variant was identified as rs138349 (chr22:41249522:A > G, PP = 21.9%). For plotting purposes, we assigned a single *p*-value for gene-level significance based on Bonferroni-correction (0.05 divided by the number of

independent variants tested in fetal-like iPSC-PPC; horizontal line). We note that this *p*-value does not reflect the thresholds used to define a significant eQTL in the original adult studies[10,11]. Therefore, while the *ST13* eQTL in adult islets in panel **c** is below the horizontal line, it had an FDR < 1% in the original study[11]. In each panel, the X-axis represents variant positions while the Y-axis either shows the −log10(eQTL *p*-value) for the associations between the genotype of the tested variants and gene expression or the −log10(GWAS *p*-value) for the associations between the tested variants and the GWAS trait. For GWAS significance, we used −log10(5 × 10⁻⁸). Red vertical lines indicate the positions of the lead candidate causal variants underlying the colocalization based on maximum PP. For loci that colocalized with multiple GWAS traits, we used the credible set that yielded the smallest number of variants to plot the "PP" fine-mapping panel.

integrating fetal-like iPSC-PPC eQTLs can help resolve certain GWAS loci that cannot be resolved using only adult datasets.

Below, we demonstrate the application of our spatiotemporally informed eQTL resource by providing a detailed description of eight GWAS loci. We propose potential causal mechanisms and offer insights into their spatiotemporal contexts.

**Singleton $e_g$QTLs.** Here, we elucidate probable causal mechanisms for two GWAS loci associated with FG levels and T1D-risk that colocalized with iPSC-PPC-unique singleton $e_g$QTLs.

**chr8:80998464-81093464 and TPD52 (iPSC-PPC-unique single-ton).** We found that in the chr8:80998464-81093464 locus, a GWAS signal associated with FG levels colocalized with a fetal-like iPSC-PPC-unique singleton $e_g$QTL for *TPD52*, also known as tumor protein D52 (effect size = −0.99, PP.H4 = 91.7%) (Fig. 6a, Supplementary Fig. 14A, Supplementary Data 13). The reported causal variant underlying this GWAS signal is rs12541643[2]; however, colocalization with our eQTLs identified rs12549167 (chr8:81078464:C > T, PP = 33.9%, r² = 0.317 with rs12541643, Supplementary Data 14) as the most likely candidate causal variant underlying both *TPD52* expression in fetal-like iPSC-PPC and

FG association. TPD52 is a direct interactor with the AMP-activated protein kinase (AMPK) and negatively affects AMPK signaling. AMPK controls a wide range of metabolic processes and is responsible for maintaining cellular energy homeostasis particularly in tissues associated with obesity, insulin resistance, T2D, and cancer such as muscle, liver, hypothalamus, and the pancreas[62–65]. Dysregulation of AMPK has also been associated with developmental defects in which AMPK activation can lead to fetal malformation[66]. Our findings suggest that decreased expression of *TPD52* during development may influence changes in glucose metabolism and therefore fasting glucose levels during adult stage.

**chr9:4232083-4352083 and CDC37L1-DT (iPSC-PPC-unique singleton).** We found that the well-known *GLIS3* GWAS locus associated with FG and T1D-risk[67,68] colocalized with a fetal-like iPSC-PPC-unique singleton $e_g$QTL for the lncRNA CDC37L1 divergent transcript (*CDC37L1-DT*; effect size = 1.46; PP.H4 for FG and T1D = 92.4% and 91.2%, respectively, Fig. 6b, Supplementary Fig. 14B, Supplementary Data 13). Consistent with previous studies[67,68,69], we identified rs10758593 (chr9:4292083:G > A, PP = 79.2%) as the lead candidate causal variant underlying both eQTL and GWAS associations. Because

*GLIS3* plays a critical role in pancreatic beta cell development and function[70,71], it has often been reported as the susceptibility gene for this signal, however it remains unclear what effects rs10758593 has on *GLIS3* expression. Our analysis suggests that another potential gene target of rs10758593, specifically during pancreas development, is *CDC37L1-DT*. While the molecular function of *CDC37L1-DT* is unknown, the gene has been associated with 9p duplication in neurodevelopmental disorders[72]. Furthermore, a recent study observed a significant association between the rs10758593 risk allele and birth weight, indicating a development role played by this locus[73]. Although additional studies are needed to understand the function of *CDC37L1-DT* during pancreas development and in T1D pathology, our analysis indicates that *CDC37L1-DT* may be another candidate susceptibility gene for the variants in the *GLIS3* locus. Assessment of *GLIS3* e_gQTLs in the three pancreatic tissues showed that there was no overlap between the e_gQTLs and GWAS variants (Supplementary Fig. 15A).

**Combinatorial e_gQTLs.** Below, we describe two GWAS intervals associated with cholesterol, LDL direct levels, and T1D. We show that the GWAS variants colocalized with combinatorial e_gQTLs, indicating that multiple genes, and possibly multiple developmental stages of the pancreas, may be involved in trait predisposition.

**chr22:41049522-41449522 and ADSL and ST13 (fetal-adult combinatorial).** We found that the GWAS signals associated with cholesterol and LDL direct levels in the chr22:41049522-41449522 locus colocalized with a "fetal-adult" e_gQTL module (module ID: GE_22_63, category E) (Fig. 6c, Supplementary Fig. 14C, D, Supplementary Data 13). The module was associated with different eGenes between fetal-like iPSC-PPC and both adult pancreatic tissues, in which the GWAS variants were associated with *ADSL* expression in iPSC-PPC (effect size = 0.78) but *ST13* expression in adult whole pancreas (effect size = 0.27) and adult islets (effect size = −0.15, weakly associated). Infants born with ADSL (adenylosuccinate lyase) deficiency suffer from impaired glucose and lipid metabolism, while ST13, also known as Hsc70-interacting protein, is involved in lipid metabolism[74,75]. Overexpression of *ST13* was found to result in disordered lipid metabolism in chronic pancreatitis[74]. Although *ST13* was reported to be the candidate causal gene for this locus[76], we determined that the underlying variants may also affect *ADSL* expression but specifically during early pancreas development. Congruent with the previous study[76], our colocalization identified rs138349 (chr22:41249522:A > G, PP = 21.9% for cholesterol and 20.9% for LDL) as the lead candidate causal variant for the e_gQTLs and both cholesterol and LDL GWAS associations (Supplementary Data 14). Altogether, annotation of the chr22:41049522-41449522 GWAS locus using our pancreatic eQTL resource suggests that altered expression of *ADSL* during pancreas development and *ST13* in adult pancreatic tissues may contribute to changes in cholesterol and LDL direct levels in adult. Additional studies are required to understand the degree to which *ADSL* and *ST13* are causal for cholesterol and LDL direct levels.

**chr10:90001035-90066035 and PTEN and LIPJ (adult whole pancreas-unique combinatorial).** We found a T1D-risk signal in the chr10:90001035-90066035 locus that colocalized with an "adult whole pancreas-unique" e_gQTL module (module ID: GE_10_35) associated with *PTEN* and *LIPJ* expression in adult whole pancreas (effect size = 0.48 and 0.49, respectively) (Supplementary Fig. 14E, Supplementary Fig. 15B, Supplementary Data 13). Colocalization identified the distal regulatory variant rs7068821 (chr10:90051035:G > T; PP = 85.5%) as the most likely candidate causal variant (Supplementary Data 14), which is in LD with the reported index SNP rs10509540 (r² = 0.876) in the GWAS catalogue. While *RNLS* was reported to be the susceptibility gene for this locus[77], our analysis suggests that both *PTEN* and *LIPJ* may be candidate causal genes for this locus. Previous studies have shown

that knockout of pancreas-specific *PTEN* (PPKO) in mice resulted in enlarged pancreas and elevated proliferation of acinar cells. PPKO mice also exhibited hypoglycemia, hypoinsulinemia, and altered amino metabolism[78]. LIPJ encodes the lipase family member J and is involved in lipid metabolism[79]. Our findings provide additional biological insight into the chr10:900001035-90066035 T1D locus and support previous studies suggesting a potential causal role of the adult whole pancreas in T1D pathogenesis[3,73].

**Singleton and combinatorial e_ASQTLs.** Here, we illustrate three examples of putative causal variants involved in alternative splicing uniquely in the fetal-like iPSC-PPC. Long-noncoding RNAs (lncRNAs) have previously been shown to play important roles in both pancreas development and diseases[80]. One of our examples includes a lncRNA while two involve protein-coding genes.

**chr14:101286447-101326447 and MEG3 (iPSC-PPC-unique singleton).** The chr14:101286447-101326447 is a well-known GWAS locus associated with T1D and has been reported to affect the lncRNA maternally expressed gene 3 (*MEG3*). While the role of *MEG3* in T1D and T2D pathogenesis has been extensively studied[81–83], the genetic mechanism by which this locus affects *MEG3* expression and therefore, T1D-risk is not well understood. Using our pancreatic eQTL resource, we found that the GWAS signal colocalized with a fetal-like iPSC-PPC-unique singleton e_ASQTL for a *MEG3* isoform (ENST00000522618, PP.H4 = 98%, effect size = 1.3, Fig. 7a, Supplementary Fig. 16A, Supplementary Data 13). Colocalization with the *MEG3* e_ASQTL identified rs56994090 (chr14:101306447:T > C, PP = 100%) as the most likely candidate causal variant, which is concordant with the findings of a previous GWAS study[84] (Supplementary Data 14). Of note, rs56994090 is in strong LD with an indel (rs34552516), which was previously identified as a candidate causal variant.[85] Given that rs56994090 is located in the novel intron enhancer of *MEG3*[81], we hypothesize that alternative splicing of *MEG3* may alter the enhancer's regulatory function, as previously observed in other lncRNAs[86], and thereby, affect T1D-risk. Altogether, our findings describe a potential causal mechanism for the T1D-risk locus involving differential alternative splicing of *MEG3* specifically during pancreas development.

**chr16:684685635-68855635 and CDH3 (iPSC-PPC-unique singleton).** We determined a known GWAS signal in the chr16:684685635-68855635 locus associated with HbA1c levels[87] colocalized with a fetal-like iPSC-PPC-unique singleton e_ASQTL for the P-cadherin 3 (*CDH3*) isoform ENST00000429102 (effect size = −1.6, PP.H4 = 83.1%) (Fig. 7b, Supplementary Fig. 16B, Supplementary Data 13). Colocalization using the e_ASQTL identified intronic variant rs72785165 (chr16:68755635:T > A, PP = 6.8%) as the most likely candidate causal variant (Supplementary Data 14), which is in high LD with the reported GWAS SNP (rs4783565, r² = 0.88)[87]. While it remains unclear how alternative splicing of *CDH3* affects HbA1c levels, studies have shown that chimeric proteins made of cadherin ectodomains, including the P-cadherin CDH3, are important for proper insulin secretion by pancreatic beta cells[88]. Based on our findings, we hypothesize that differential isoform usage of *CDH3* during pancreas development may influence glucose control and therefore, HbA1c levels, in adults.

**chr13:30956642-31116642 and HMGB1 (iPSC-PPC-unique combinatorial).** The GWAS signals associated with T2D and BMI in the chr13:30956642-31116642 locus[52,56,89–91] colocalized with the iPSC-PPC-unique e_ASQTL module (module ID: AS_13_2) associated with three *HMGB1* isoforms: ENST00000326004, ENST00000399494, ENST00000339872, and (effect size = 2.16, −2.26, and −0.85, respectively; HMGB1.1, HGMB1.2, and HMGB1.3, respectively) (Fig. 7c, Supplementary Fig. 16C–E, Supplementary Data 13). Our colocalization

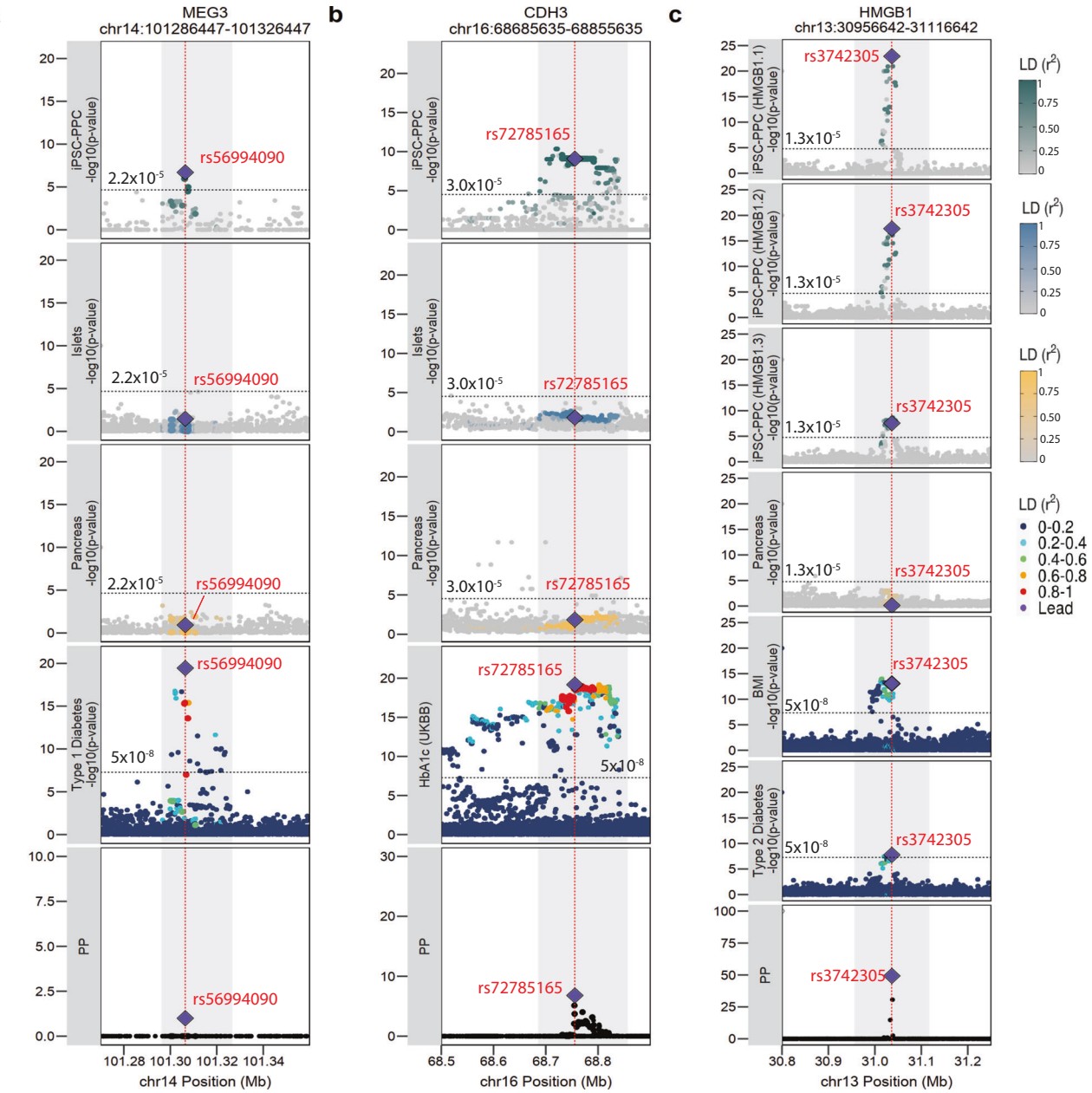

**Fig. 7 | Pancreatic GWAS associations with fetal-specific alternative splicing.**
**a** T1D-risk locus colocalized with a fetal-like iPSC-PPC-unique singleton e$_{AS}$QTL for
*MEG3* with the predicted causal variant rs56994090 (chr14:101306447:T > C, PP =
100%). **b** GWAS locus associated with HbA1c colocalized with an iPSC-PPC-unique
singleton e$_{AS}$QTL for *CDH3* with the predicted causal variant rs72785165
(chr16:68755635:T > A, PP = 6.8%). **c** GWAS locus associated with T2D-risk and BMI
colocalized with an iPSC-PPC-unique e$_{AS}$QTL module (AS_13_2) for differential
usage of three *HMGB1* isoforms with a predicted causal variant rs3742305
(chr13:31036642:C > G, PP = 49.3%). In each panel, the X-axis represents variant
positions while the Y-axis either shows the −log10(eQTL *p*-value) for the

associations between the genotype of the tested variants and gene expression or
the −log10(GWAS *p*-value) for the associations between the tested variants and the
GWAS trait. For GWAS significance, we used −log10(5 × 10$^{-8}$). For eQTL significance,
we used a single *p*-value for gene-level significance after Bonferroni-correction
(0.05/the number of independent variants tested in fetal-like iPSC-PPC; horizontal
line). Red vertical lines indicate the positions of the lead candidate causal variants
underlying the colocalization based on maximum PP. For loci that colocalized with
multiple GWAS traits, we used the credible set that yielded the smallest number of
variants to plot the "PP" fine-mapping panel.

identified rs3742305 (chr13:31036642:C > G, PP = 49.3%) as a lead
candidate causal variant underlying this locus, in which the risk allele
(G) was associated with increased usage of ENST00000326004 and
decreased usages of ENST0000339872 and ENST00000399494
(Supplementary Fig. 16C–E, Supplementary Data 7, Supplementary
Data 14). While a previous study[90] also reported *HMGB1* as the sus-
ceptibility gene, the precise mechanism by which rs3742305 affected
*HMGB1* expression was unclear. HMGB1, also known as high-mobility
group box 1, is an important mediator for regulating gene expression
during both developmental and adult stages of life. Deletion of *HMGB1*

disrupts cell growth and causes lethal hypoglycemia in mouse
pups[92,93]. In T2D, *HMGB1* promotes obesity-induced adipose inflam-
mation, insulin resistance, and islet dysfunction[91]. Our results suggest
that differential usage of *HMGB1* isoforms during pancreas develop-
ment may affect adult risk of developing obesity and/or T2D.

Altogether, our findings demonstrate the value of our pancreatic
eQTL resource to annotate GWAS risk variants with fetal-like and adult
temporal and spatial regulatory information. We show that some
causal regulatory variants underlying disease-associated signals may
influence adult traits by modulating the expression of genes in early

development, while in other cases, they may display regulatory plasticity and exert their effects by modulating the expression of multiple different genes in fetal-like and adult pancreatic stages. Further, we identified an association between whole pancreas and T1D, supporting a potential role of this tissue in diabetes pathogenesis[3].

## Discussion

In this study, we leveraged one of the most well-characterized iPSC cohorts comprising >100 genotyped individuals to derive pancreatic progenitor cells and generate a comprehensive eQTL resource for examining genetic associations with gene expression and isoform usage in fetal-like pancreatic cells. We discovered 8665 eQTLs in the fetal-like iPSC-PPCs and showed that 60% of eGenes were associated with regulatory variation uniquely active during pancreas development. For the eGenes that were shared with adult, ~12% were associated with different genomic loci, indicating that different regulatory elements may modulate the same gene in fetal-like and adult pancreatic stages. We further identified regulatory variants that displayed early pancreas development-unique function, of which 1805 were uniquely active in only iPSC-PPC and 1043 were active in both developmental and adult contexts but exhibited regulatory plasticity in the genes they regulate. These results concur with previous studies showing that the genetic regulatory landscape changes between fetal tissues and their adult counterparts[94–96], and therefore, highlights the importance of assessing variant function in both fetal and adult tissue contexts. Furthermore, it is widely known that tight regulation of genes during development is essential[97], and our study reflects this in our findings that the majority of developmental-unique eQTLs were restricted to a single eGene.

Finally, we highlighted examples of GWAS associations for which we utilized our spatiotemporally informed eQTL resource to characterize causal risk mechanisms underlying adult pancreatic disease. We showed that some causal regulatory variants underlying GWAS signals identified in the fetal-like iPSC-PPCs modulate the expression of genes in early development, while others may exert their effects by modulating the expression of multiple different genes across fetal-like and adult pancreatic stages. Of note, many of the fetal-unique regulatory variants underlying the GWAS signals were e$_{AS}$QTLs, which is consistent with alternative splicing playing a key role in developing tissues[59–61,98]. Hence, we believe that contribution of alternative splicing differences during fetal pancreas development to complex traits warrants further investigation.

We offer limitations in our study and potential future directions for the field at large. We believe that studies using larger sample sizes are needed to identify additional associations between genetic variation and gene expression in fetal samples. Our eQTL mapping in fetal-like iPSC-PPC was conducted on much fewer samples compared to the two adult studies that each used ~300−400 samples, rendering our dataset underpowered and not able to capture additional eQTL associations that could be shared with the adult pancreatic tissues. Therefore, several eQTLs we annotated as adult islet-unique or whole pancreas-unique may in reality be shared with fetal pancreas. Further, power differences between the studies may also cause the observed results where there were many singleton eQTLs observed in a single tissue. On the other hand, the eQTLs we annotated as iPSC-PPC-unique may less likely to be shared, as the signals in the adult datasets are better powered and therefore sufficient for comparing against iPSC-PPC signals. Additionally, with the rapid generation of eQTL datasets from different tissue contexts[8,9,10,13], the development and application of artificial intelligence and machine learning as ways to identify shared eQTL associations between multiple tissues will be extremely useful. While pairwise colocalization and network analysis can identify shared eQTL regulatory loci across a handful of tissues, machine learning approaches could scale these analyses across spatiotemporal contexts of all tissues, thereby providing valuable insights into

regulatory elements that are exclusive to a specific context and also those that exhibit regulatory plasticity across multiple contexts.

In summary, our study provides a valuable resource for discovering causal regulatory mechanisms underlying pancreatic traits and disease across developmental and adult timepoints of the pancreas. We reveal that disease variants may either display temporal-specificity in which they affect gene expression specifically in one timepoint, or regulatory plasticity, in which they affect gene expression in multiple timepoints but affect different genes. Our findings lay the groundwork for future employment of development contexts for the characterization of disease-associated variants.

## Methods

### Subject information

We used iPSC lines from 106 individuals recruited as part of the iPSCORE project (Supplementary Data 1). There were 53 individuals belonging to 19 families composed of two or more subjects (range: 2-6). Each subject was assigned an iPSCORE_ID (i.e., iPSCORE 4_1), where "4" indicates the family number and "1" indicates the individual number, and a 128-bit universal unique identifier (UUID). The 106 individuals included 68 females and 38 males with ages ranging from 15 to 88 years old at the time of enrollment. Recruitment of these individuals was approved by the Institutional Review Boards of the University of California, San Diego, and The Salk Institute (project no. 110776ZF).

### WGS data

Whole-genome sequencing data for the 106 iPSCORE individuals were downloaded from dbGaP (phs001325.v3) as a VCF file[41]. We retained variants with MAF > 5% across all 273 individuals in the iPSCORE resource, that were in Hardy-Weinberg equilibrium ($p > 10^{-6}$), and that were within 500 Kb of the expressed gene's body coordinates. Specifically, we expanded the coordinates of each of the 16,464 expressed autosomal genes (500 Kb upstream and downstream) and extracted all variants within these regions using *bcftools view* with parameters *--f PASS -q 0.05:minor*[99]. Next, we normalized indels and split multi-allelic variants using *bcftools norm -m-* and removed variants that were genotyped in fewer than 99% of samples using *bcftools filter -i 'F_PASS(GT! ="mis") > 0.99'*[99]. Finally, we converted the resulting VCF files to text using *bcftools query*[99] and converted the genotypes from character strings (0/0, 0/1, and 1/1) to numeric (0, 0.5, and 1, respectively). This resulted in 6,593,484 total variants used for eQTL mapping.

### iPSC generation

Generation of the 106 iPSC lines has previously been described in detail[39]. Briefly, cultures of primary dermal fibroblast cells were generated from a punch biopsy tissue[100], infected with the Cytotune Sendai virus (Life Technologies) per manufacturer's protocol to initiate reprogramming. Emerging iPSC colonies were manually picked after Day 21 and maintained on Matrigel (BD Corning) with mTeSR1 medium (Stem Cell Technologies). Multiple independently established iPSC clones (i.e. referred to as lines) were derived from each individual. Many of the iPSC lines were evaluated by flow cytometry for expression of two pluripotent markers: Tra-1-81 (Alexa Fluor 488 anti-human, Biolegend) and SSEA-4 (PE anti-human, Biolegend)[39]. Pluripotency was also examined using PluriTest-RNAseq[39]. This iPSCORE resource was established as part of the Next Generation Consortium of the National Heart, Lung and Blood Institute and is available to researchers through the biorepository at WiCell Research Institute (www.wicell.org; NHLBI Next Gen Collection). For-profit organizations can contact the corresponding author directly to discuss line availability.

### Pancreatic progenitor differentiation

We performed pancreatic progenitor cell (PPC) differentiation on each of the 106 iPSC lines. One iPSC line was differentiated twice giving a total of 107 differentiations. Each differentiation was assigned a 128-bit

universally unique identifier (UUID), and a unique differentiation ID (UDID; "PPCXXX"), where "XXX" represents a numeric integer (Supplementary Data 2).

**Differentiation protocol.** The iPSC lines were differentiated into PPCs using the STEMdiff™ Pancreatic Progenitor Kit (StemCell Technologies) protocol with minor modifications. Briefly, iPSC lines were thawed into mTeSR1 medium containing 10 μM Y-27632 ROCK Inhibitor (Selleckchem) and plated onto one well of a 6-well plate coated with Matrigel. iPSCs were grown until they reached 80% confluency[101] and then passaged using 2 mg/ml solution of Dispase II (ThermoFisher Scientific) onto three wells of a 6-well plate (ratio 1:3). To expand the iPSC cells for differentiation, iPSCs were passaged a second time onto six wells of a 6-well plate (ratio 1:2). When the iPSCs reached 80% confluency, cells were dissociated into single cells using Accutase (Innovative Cell Technologies Inc.) and resuspended at a concentration of $1.85 \times 10^6$ cells/ml in mTeSR medium containing 10 μM Y-27632 ROCK inhibitor. Cells were then plated onto six wells of a 6-well plate and grown for approximately 16 to 20 hours to achieve a uniform monolayer of 90–95% confluence ($3.7 \times 10^6$ cells/well; about $3.9 \times 10^5$ cells/cm²). Differentiation of the iPSC monolayers was initiated by the addition of the STEMdiff™ Stage Endoderm Basal medium supplemented with Supplement MR and Supplement CJ (2 ml/well) (Day 1, D1). The following media changes were performed every 24 hours following initiation of differentiation (2 ml/well). On D2 and D3, the medium was changed to fresh STEMdiff™ Stage Endoderm Basal medium supplemented with Supplement CJ. On D4, the medium was changed to STEMdiff™ Pancreatic Stage 2-4 Basal medium supplemented with Supplement 2 A and Supplement 2B. On D5 and D6, the medium was changed to STEMdiff™ Pancreatic Stage 2-4 Basal medium supplemented with Supplement 2B. From D7 to D9, the medium was changed to STEMdiff™ Pancreatic Stage 2-4 Basal medium supplemented with Supplement 3. From D10 to D14, the medium was changed to STEMdiff™ Pancreatic Stage 2-4 Basal medium supplemented with Supplement 4. On D15, cells were dissociated with Accutase and then collected, counted, and processed for data generation. iPSC-PPC cells were cryopreserved in CryoStor® CS10 (StemCell Technologies).

**iPSC-PPC differentiation efficiency.** To evaluate the efficiency of iPSC-PPC differentiation, we performed flow cytometry on two pancreatic precursor markers, PDX1 and NKX6-1. Specifically, at least $2 \times 10^6$ cells were fixed and permeabilized using the Fixation/Permeabilized Solution Kit with BD GolgiStop TM (BD Biosciences) following the manufacturer's recommendations. Cells were resuspended in 1x BD Perm/Wash TM Buffer at a concentration of $1 \times 10^7$ cells/ml. For each flow cytometry staining, $2.5 \times 10^5$ cells were stained for 75 minutes at room temperature with PE Mouse anti-PDX1 Clone-658A5 (BD Biosciences; Catalog no. 562161; 1:10) and Alexa Fluor® 647 Mouse anti-NKX6.1 Clone R11-560 (BD Bioscience; Catalog no. 563338; 1:10), or with the appropriate class control antibodies: PE Mouse anti-IgG1 κ R-PE Clone MOPC-21 (BD Biosciences; Catalog no. 559320) and Alexa Fluor® 647 Mouse anti IgG1 κ Isotype Clone MOPC-21 (BD Biosciences; Catalog no. 557732). PE Mouse anti-PDX1 Clone-658A5 and Alexa Fluor® 647 Mouse anti-NKX6.1 Clone R11-560 were validated by the manufacturer to bind to mouse and human PDX-1 and NKX6-1, respectively. Stained cells were washed three times, resuspended in PBS containing 1% BSA and 1% formaldehyde, and immediately analyzed using FACS Canto II flow cytometer (BD Biosciences). The fraction of PDX1- and NKX6-1-positive was calculated using FlowJo software version 10.4 (Supplementary Data 2).

**scRNA-seq**
To characterize the cellular composition of the fetal-like iPSC-PPC samples, we performed single-cell RNA-seq (scRNA-seq) on one iPSC line (from differentiation PPC034) and ten iPSC-PPC samples with varying percentages of double-positive PDX1+/NKX6-1+ cells based on flow cytometry (range: 9.4–91.7%) (Supplementary Fig. 2, Supplementary Fig. 3, Supplementary Data 2). Because bulk RNA-seq was generated on cryopreserved cells, we sought to also examine whether cell cryopreservation affects gene expression estimates using scRNA-seq. Therefore, we included both freshly prepared (i.e., not frozen and processed immediately after differentiation) and cryopreserved cells for four iPSC-PPC samples (PPC029, PPC027, PPC023, PPC034; Supplementary Data 2) for scRNA-seq processing.

**Sample collection.** Fresh cells from the iPSC line and seven iPSC-PPC samples were captured individually at D15. Cells from four of these same iPSC-PPC samples that had been cryopreserved were pooled and captured immediately after thawing (RNA_Pool_1). Cells from an additional three iPSC-PPC samples were captured only after cryopreservation (RNA_Pool_2) (Supplementary Data 2).

**Library preparation and sequencing.** All single cells were captured using the 10X Chromium controller (10X Genomics) according to the manufacturer's specifications and manual (Manual CG000183, Rev A). Cells from each scRNA-seq sample (one iPSC, seven fresh iPSC-PPCs, RNA_Pool_1, and RNA_Pool_2) were loaded each onto an individual lane of a Chromium Single Cell Chip B. Libraries were generated using Chromium Single Cell 3' Library Gel Bead Kit v3 (10X Genomics) following manufacturer's manual with small modifications. Specifically, the purified cDNA was eluted in 24 μl of Buffer EB, half of which was used for the subsequent step of the library construction. cDNA was amplified for 10 cycles and libraries were amplified for 8 cycles. All libraries were sequenced on a HiSeq 4000 using custom programs (fresh: 28-8-175 Pair End and cryopreserved: 28-8-98 Pair End). Specifically, eight libraries generated from fresh samples (one iPSC and seven iPSC-PPC samples) were pooled together and loaded evenly onto eight lanes and sequenced to an average depth of 163 million reads. The two libraries from seven cryopreserved lines (RNA_Pool_1 and RNA_Pool_2) were each sequenced on an individual lane to an average depth of 265 million reads. In total, we captured 99,819 cells. We observed highly correlated cell type proportions between fresh and cryopreserved iPSC-PPC samples (Supplementary Fig. 8).

**scRNA-seq alignment.** We obtained FASTQ files for the ten scRNA-seq samples (one iPSC, seven fresh iPSC-PPCs, RNA_Pool_1, and RNA_Pool_2) (Supplementary Data 2) and used CellRanger V6.0.1 (https://support.10xgenomics.com/) with default parameters and GENCODE version 34 hg19[102] gene annotations to generate single-cell gene counts and BAM files for each of the ten samples.

**Dataset integration and quality control.** We processed the single-cell gene counts by first aggregating the iPSC and seven fresh iPSC-PPC samples using the *aggr* function on CellRanger V6.0.1 with normalization = F. Then, we integrated the aggregated dataset ("aggr") with the two pools of cryopreserved samples (RNA_Pool_1 and RNA_Pool_2) using the standard integration workflow described in Seurat (version 3.2; https://satijalab.org/seurat/archive/v3.2/integration.html). Specifically, for each dataset (aggr, RNA_Pool_1, and RNA_Pool_2), we log-normalized the gene counts using *NormalizeData* (default parameters) then used *FindVariableFeatures* with *selection.method = "vst"*, *nfeatures = 2000*, and *dispersion.cutoff = c(0.5, Inf)* to identify the top 2000 most variable genes in each dataset. We then used *FindIntegrationAnchors* and *IntegrateData* with *dims = 1:30* to integrate the three datasets. We scaled the integrated data with *ScaleData*, performed principal component analysis with *RunPCA* for *npcs = 30*, and processed for UMAP visualization (*RunUMAP* with *reduction = "pca"* and *dims = 1:30*). Clusters were identified using *FindClusters* with default parameters.

To remove low-quality cells, we examined the distribution of the number of genes per cell and the percentage of reads mapping to the mitochondrial chromosome (chrM) in each cluster. We removed the cluster (11,677 cells) with fewer than 500 genes per cell and more than 50% of the reads mapping to chrM. We re-processed the filtered data (*ScaleData, RunPCA, FindClusters, RunUMAP*) and removed another cluster of cells that had the lowest median number of expressed genes (723 versus 2775) and highest median fraction of mitochondrial reads (34.0% versus 8.39%). After this second filtering step, we retained 84,258 cells.

**Demultiplexing sample identity.** We used Demuxlet[103] to assign pooled cryopreserved cells in RNA_Pool_1 and RNA_Pool_2 (19,136 cells in total) to the correct iPSC-PPC sample. Specifically, we provided CellRanger BAM files and a VCF file containing genotypes for biallelic SNVs located at UTR and exon regions on autosomes as annotated by GENCODE version 34 hg19[102]. We excluded 33 cells that were incorrectly assigned to samples not associated with the pooled sample (i.e., cells from RNA_POOL_1 were predicted to be from other samples not in RNA_Pool_1). 84,225 cells remained for downstream analyses (Supplementary Data 3).

**Annotation of cell type clusters.** We annotated the scRNA-seq clusters by first clustering with three different resolutions (0.5, 0.08, and 0.1) (Supplementary Figs. 4–6). We selected resolution = 0.08 because it best captured the expected iPSC-PPC cell types based on each cluster's expression for the following gene markers: *POU5F1* (iPSC), *COL1A1, COL1A2* (mesendoderm) *AFP, APOA* (early definitive endoderm), *GATA4, GATA6, PDX1* (early PPC), *PDX1, NKX6-1* (late PPC), *PAX6, CHGA, INS, GCG, SST* (endocrine), and *FLT1* (early ductal). We validated our annotations by comparing the iPSC-PPC clusters to those identified from scRNA-seq of ESC-PPC samples over 4 different stages of differentiation[27] (GSE114412): Stage 3 (Day 6; 7982 cells), Stage 4 (Day 13; 6960 cells), Stage 5 (Day 18; 4,193 cells), and Stage 6 (Day 25; 5186 cells). Specifically, we compared the expression patterns of the gene markers between the clusters using z-normalized mean expression computed on cells expressing at least 1% of maximal expression for the gene, as described in the reference study[27]. Metadata containing single cell annotations are reported in Supplementary Data 3.

**Differentially expressed genes.** To identify differentially expressed genes for each iPSC-PPC cluster, we used the *FindAllMarkers* function in Seurat[104] with *logfc.threshold = 0.01* and *min.pct = 0.01*. *P*-values were automatically adjusted by Seurat using Bonferroni correction, and genes with adjusted *p*-values ≤ 0.05 were considered differentially expressed (Supplementary Data 4).

**Bulk RNA-seq**
**Library preparation and sequencing.** RNA was isolated from total-cell lysates using the Quick-RNA™ MiniPrep Kit (Zymo Research) with on-column DNAse treatments. RNA was eluted in 48 μl RNAse-free water and analyzed on a TapeStation (Agilent) to determine sample integrity. All iPSC-PPC samples had RNA integrity number (RIN) values over 9. Illumina TruSeq Stranded mRNA libraries were prepared according to the manufacturer's instructions and sequenced on NovaSeq6000 for 101 bp paired-end sequencing. All samples except five were sequenced twice to obtain sufficient number of reads.

**Data processing and quality control.** FASTQ files were obtained for all 107 iPSC-PPC samples and processed using a similar pipeline described in our previous studies[12,41,105]. Specifically, RNA-seq reads were aligned with STAR (2.7.3)[106] to the hg19 reference using GEN-CODE version 34 hg19[102] splice junctions with default alignment parameters and the following adjustments: *-outFilterMultimapNmax 20, -outFilterMismatchNmax 999, -alignIntronMin 20, -alignIntronMax*

*1000000, -alignMatesGapMax 1000000*. BAM files were sorted by coordinates, and duplicate reads were marked using Samtools (1.9.0)[99]. RNA-seq QC metrics were calculated using Samtools (1.9.0) flagstat[99], Samtools (1.9.0) idxstats[99], and Picard (2.20.1) CollectRnaSeqMetrics (https://broadinstitute.github.io/picard/). Across all 107 iPSC-PPC samples, the total read depth ranged from 32.3 M to 160.4 M (mean = 70.7), the median percentage of intergenic bases was 3.31%, the median percentage of mRNA bases was 92.1%, and the median percentage of duplicate reads was 22.2% (Supplementary Data 2).

**Sample identity.** We confirmed sample identity by obtaining common bi-allelic and exonic variants from the 1000 Genomes Phase 3 panel[107] with minor allele frequencies between 45% and 55% and predicting their genotypes in the 107 bulk RNA-seq samples using *mpileup* and *call* functions in BCFtools (1.9.0)[99,108]. Then, we used the *genome* command in plink[109] to estimate the identity-by-state (IBS) between each pair of bulk RNA-seq and WGS samples. All RNA-seq samples were correctly matched to the subject with PI_HAT > 0.95 (Supplementary Data 2).

**Quantification of gene expression and relative isoform usage.** We calculated TPM and estimated relative isoform usage for each gene in each RNA-seq sample using RSEM (version 1.2.20)[110] with the following options *−seed 3272015 −estimate-rspd −paired-end −forward-prob*. To identify expressed autosomal genes and isoforms to use for eQTL analyses, we used the same approach previously described[12]. Briefly, autosomal genes were considered expressed if TPM ≥ 1 in at least 10% of samples. To identify expressed isoforms, we required that isoforms had TPM ≥ 1 and usage ≥ 10% in at least 10% of samples and corresponded to expressed genes with at least two expressed isoforms. In total, 16,464 autosomal genes were used for $e_gQTL$ analysis, and 29,871 autosomal isoforms corresponding to 9624 genes were used for $e_iQTL$ analysis. We quantile-normalized TPM and isoform usage across all 107 samples using the *normalize.quantiles* (preprocessCore) and *qnorm* functions in R (version 4.2.1) to obtain a mean expression = 0 and standard deviation = 1.

**Inferring pseudotime using Monocle.** We obtained FASTQ files for 213 iPSCs[39,41] (phs000924), 176 adult whole pancreas[8] (phs000424), and 87 adult islets[42] (GSE50398), and processed the data using the same pipeline described above to obtain TPM counts for each gene per sample. We then used Monocle (http://cole-trapnell-lab.github.io/monocle-release/docs/#constructing-single-cell-trajectories)[111] to infer the pseudotime on all of the RNA-seq samples, including the 107 iPSC-PPCs. Following the standard workflow under "Constructing Single Cell Trajectories" in the Monocle tutorial, we provided TPM counts for all overlapping autosomal expressed genes in the four tissues as input. Then, we identified differentially expressed genes using *differentialGeneTest*, ordered them (*setOrderingFilter*), and performed dimension reduction analysis using *reduceDimension* with *max_components = 2* and *method = "DDRTree"*. Pseudotime was calculated by rooting time (pseudotime = 0) in the 213 iPSC-PPCs using the *GM_state* and *orderCells* functions provided in the tutorial (Supplementary Data 6).

**PCA analysis with iPSCs, adult whole pancreas, and adult islets.** We obtained TPM counts (described above) for the 213 iPSCs[39,41], 176 adult whole pancreas[8], 87 adult islets[42], and the 107 iPSC-PPCs and performed PCA analysis on the 2000 most variable genes across the samples using *prcomp* in R (version 4.2.1) with *scale = T* and *center = T*. We observed that the PC clusters corresponded to the iPSCs and each of the three pancreatic tissue types: iPSC-PPC, adult islets, and adult whole pancreas (Supplementary Fig. 9, Supplementary Data 6).

**Cellular deconvolution.** For each of the eight cell types in scRNA-seq, we selected the top 200 most differentially expressed genes that were unique to the cell type (i.e., not expressed in the other cell types). Replicating late PPCs and late PPCs had many overlapping expressed genes so fewer ($n = 16$ and 164, respectively) were selected. We obtained the average expression of the signature genes for each cell type using *AverageExpression* in Seurat and provided it as input into CIBERSORTx[112] (https://cibersortx.stanford.edu/) along with bulk TPM matrix. Batch correction and quantile normalization were both disabled. We ran CIBERSORTx[112] deconvolution on absolute mode with at least 100 permutations. The predicted fraction of late PPCs and replicating late PPCs were compared to FACS measurements of double-positive PDX-1⁺/NKX6-1⁺ cells (Supplementary Fig. 7B; Supplementary Data 5).

## eQTL analysis

To investigate the effects of genetic variation on gene expression in iPSC-PPCs, we performed an expression quantitative trait loci (eQTL) analysis on gene expression and isoform usage. The eQTLs associated with gene expression were defined as $e_gQTLs$ while those associated with relative isoform usage were defined as $e_iQTLs$.

**Covariates for eQTL mapping.** We included the following as covariates for eQTL mapping of both gene expression and isoform usage: 1) sex; 2) normalized number of RNA-seq reads; 3) percent of reads that mapped to autosome or sex chromosomes (labeled as "uniquely_mapped_reads_canonical_chromosomes" in Supplementary Data 2); 4) percent of reads mapped to mitochondrial chromosome; 5) 20 genotype principal components to account for global ancestry; 6) 20 PEER factors to account for transcriptome variability; and 7) kinship matrix to account for genetic relatedness between samples. Kinship matrix is provided in Supplementary Data 1, and the covariates are available in Supplementary Data 2.

**Genotype principal component analysis (PCA).** Global ancestry was estimated using the genotypes of the 439,461 common variants with minor allele frequency (MAF) between 45 and 55% in the 1000 Genomes Phase 3 Panel[108]. We merged the VCF files for the 106 iPSCORE subjects and the 2504 subjects in the 1000 Genomes[108] and performed a PCA analysis using *plink --pca*[107] (Supplementary Fig. 1A). The top 20 principal components were used as covariates in the eQTL model to account for global ancestry and can be found in Supplementary Data 1.

**PEER factors.** We sought to determine the optimal number of PEER factors to use in the eQTL analysis that will result in maximal eGene discovery. To this end, we initially calculated PEER factors on the 10,000 expressed genes with the largest variance across all samples. To limit biases due to the expression levels of each gene, we divided the 16,464 expressed genes into ten deciles based on their average TPM, and selected 50 genes from each decile, for a total of 500 genes. We next performed eQTL analysis on each of the 500 genes using 10 to 60 PEER factors in increments of 10. While 30 PEER factors resulted in the highest percentage of eGenes (14.0%), we opted for using 20 PEER factors because the eQTL analysis had a comparable percentage of eGenes (11.8%) to GTEx tissues with similar sample sizes[10] (Supplementary Fig. 17). Although we observed variable fraction of double-positive PDX1 + /NKX6-1+ cells in the iPSC-PPC samples, we did not include this variable as a covariate because PEER factors 1 and 4 already accounted for this variability (Supplementary Fig. 18).

**Kinship matrix.** The kinship matrix was included as a random effects term to account for the genetic relatedness between individuals in our cohort. We constructed the kinship matrix using the same 439,461 variants employed above using the *–make-rel square* function in plink[107]. The kinship matrix is available in Supplementary Data 1.

**eQTL analysis.** We performed eQTL analysis using the same method described in our previous study[12]. For each expressed autosomal gene and isoform, we tested variants that were within 500 Kb of the gene body coordinates using the *bcftools query* function. To account for the genetic relatedness between the samples, we performed eQTL mapping using a linear mixed model with the *scan* function in limix (version 3.0.4)[113] that incorporates the kinship matrix as a random effects term. Specifically, eQTL mapping was implemented through the following model:

$$y_i = \beta_{ji} \cdot g_j + \sum_{n=1}^{N} \beta_n \cdot C_n + u + \epsilon_{ij}$$

Where $y_i$ is the normalized expression value for gene $i$, $\beta_{ji}$ is the effect size of genotype of SNP $j$ on gene $i$, $g_j$ is the genotype of SNP $j$, $\beta_n$ is the effect size of covariate $n$, $C_n$ is a vector of values for covariate $n$, u is the kinship matrix as a random effect, and $\epsilon$ is the error term for the association between expression of gene $i$ and genotype of SNP $j$. As described above, we used the following as covariates: 1) sex, 2) normalized number of RNA-seq reads, 3) percent of reads mapped to autosomal or sex chromosome, 4) percent of reads mapped to mitochondrial chromosome, 5) the top 20 genotype PCs (to account to global ancestry), and 6) the top 20 PEER factors (to account for confounders of expression variability), and are available in Supplementary Data 1-2.

**FDR correction.** To perform FDR correction, we used a two-step procedure described in Huang et al. [114], which first corrects at the gene level and then at the genome-wide level. First, we performed FDR correction on the *p*-values of all variants tested for each gene or isoform using eigenMT[113], which considers the LD structure of the variants. Then, we extracted the lead eQTL for each gene or isoform based on the most significant FDR-corrected *p*-value. If more than one variant had the same FDR-corrected *p*-value, we selected the one with the largest absolute effect size as the lead eQTL. For the second correction, we performed an FDR-correction on all lead variants using the Benjamini-Hochberg method (*q*-value) and considered only eQTLs with *q*-value ≤ 0.01 as significant (Supplementary Data 7).

**Conditional eQTLs.** To identify additional independent eQTLs (i.e., conditional eQTLs) for each eGene and eIsoform, we performed a stepwise regression analysis in which the genotype of the lead eQTL was included as a covariate in the model and the eQTL mapping procedure (regression and multiple test correction) was re-performed. We repeated this analysis to discover up to five additional associations for each eGene and eIsoform. Conditional eQTLs with *q*-values ≤ 0.01 were considered significant (Supplementary Data 7).

## Functional characterization of iPSC-PPC eQTLs

**Fine-mapping of eQTL associations.** To define a credible set of candidate causal variants for each eQTL association, we performed genetic fine-mapping using the *finemap.abf* function in *coloc* (version 5.1.0, R)[45]. This Bayesian method converts *p*-values of all variants tested for a specific gene to posterior probabilities (PP) of association for being the causal variant. Variants with PP 1% are available on Figshare: https://figshare.com/projects/Large-scale_eQTL_analysis_of_iPSC-PPC/156987. eQTLs not present in the table do not have variants with PP 1% (i.e., all variants were estimated to have PP < 1%).

**Genomic enrichments of $e_gQTLs$ and $e_iQTLs$.** For each independent eQTL association, we obtained candidate causal variants whose PP ≥ 5% and determined their overlap with each of the following genomic annotations using *bedtools intersect*: short splice acceptor sites (± 50 bp), long splice acceptor sites (± 100 bp), splice donor sites (± 50 bp), UTR, intron, exon, intergenic, promoters, and RNA-binding

protein binding sites (RBP-BS). RBP-BS were downloaded from a published dataset that utilized enhanced CLIP to identify binding sites of 73 RBPs[115]. We considered only binding sites with irreproducible discovery rate (IDR) threshold of 0.01, indicating that these sites were reproducible across multiple biological samples. Enrichment of candidate causal variants for genomic regions was calculated using a Fisher's Exact Test comparing the proportion of SNPs that overlap each annotation between $e_g$QTLs and $e_i$QTLs. *P*-values were corrected using the Benjamini-Hochberg method and were considered significant if their FDR-corrected *p*-value ≤ 0.05 (Fig. 1e).

**Quantification of allele-specific binding of transcription factors using GVATdb.** To annotate each candidate causal variant by their effects on transcription factor (TF) binding, we used the Genetic Variants Allelic TF Binding Database (GVATdb) to estimate the TF binding impact score associated with each variant and each of the 58 iPSC-PPC-expressed TF available on the database and with a AUPRC > 0.75 indicating a high-confidence deltaSVM model. We estimated the score using the instructions and reference files provided on the GVATdb GitHub repository (https://github.com/ren-lab/deltaSVM). The software required a list of SNPs as input along with hg19 reference files provided in the GVATdb repository. The output provides the deltaSVM score[116] for each variant-TF pair, indicating whether the variant results in a promotion ("Gain"), disruption ("Loss"), or no change ("None") in TF binding. deltaSVM scores for each variant-TF pair are available on Figshare: https://figshare.com/projects/Large-scale_eQTL_analysis_of_iPSC-PPC/156987.

**Correlation between eQTL effect size and binding affinity of transcription factors.** To determine whether $e_g$QTLs were more likely to affect TF binding compared to $e_i$QTLs, we performed a Spearman Correlation Analysis between deltaSVM score and eQTL effect size on candidate causal variants with PP ≥ 10%, 20%, 40%, 60% and 80%. We considered nominal *p*-value ≤ 0.05 as significant.

**Colocalization between iPSC-PPC gene and isoform eQTLs.** To determine the overlap of genetic variants between $e_g$QTLs and $e_i$QTLs for the same gene, we performed Bayesian colocalization using the *coloc.abf* function in *coloc* (version 5.1.0, R)[45], where each pair of signals was given a summary PP that each of the following five hypotheses was true: H0) no association was detected in both signals, H1) an association was detected only in signal 1, H2) an association was detected only in signal 2, H3) an association was detected in both signals but the underlying causal variants are different, and H4) an association was detected for both signals and the underlying causal variants are the same. We filtered the results by requiring that each colocalization used the number of overlapping variants (called "nsnps" in the *coloc.abf* output) ≥ 500. We considered two eQTL signals to be shared if the PP for H4 (called "PP.H4.abf" in *coloc.abf* output; hereafter referred to as PP.H4) ≥ 80%. Conversely, two signals were considered distinct if the PP for H3 (called "PP.H3.abf" in *coloc.abf* output; hereafter referred to as PP.H3) ≥ 80%. eQTL associations with PP.H4 < 80% and PP.H3 < 80% were due to insufficient power in one or both eQTL signals. As input into *coloc.abf*, we provided *p*-values, minor allele frequency, and sample size. All associations with PP ≥ 80% for any model are available in Supplementary Data 9.

**Genomic enrichment of overlapping $e_g$QTL and $e_i$QTL signals compared to non-overlapping.** To test the enrichment of overlapping $e_g$QTLs and $e_i$QTLs in genomic regions compared to non-overlapping signals, we determined the overlap of candidate causal variants with PP ≥ 1% in each genomic annotation using *bedtools intersect* and compared the proportion of variants overlapping each annotation against a background set of 20,000 random variants using a Fisher's Exact Test as previously described[10]. For overlapping eQTLs, we used the

candidate causal variants predicted in the *coloc.abf* output. Enrichments with nominal *p*-value < 0.05 were considered significant (Supplementary Fig. 10).

**Downloading eQTL summary statistics for adult pancreatic tissues**
We downloaded complete eQTL summary statistics for gene and exon associations for 420 adult human islets from the InSPIRE Consortium (https://zenodo.org/record/3408356)[11], and gene and splicing associations for 305 adult whole pancreas from the GTEx Data Portal for GTEx Analysis version 8[10] (https://console.cloud.google.com/storage/browser/gtex-resources). All GTEx SNPs were converted to hg19 using the UCSC liftOver Bioconductor package in R (https://www.bioconductor.org/help/workflows/liftOver/). Lead SNPs for conditional associations in the adult islets and whole pancreas datasets were downloaded from their respective studies (complete statistics were not readily available).

Due to the different types of eQTLs used in this study that are associated with changes in alternative splicing ($e_i$QTLs, exon eQTLs, and sQTLs), hereafter we refer to this collective unit as "$e_{AS}$QTLs".

**Comparing eGenes between fetal-like iPSC-PPC and adult islets**
To identify eGenes that were shared between iPSC-PPC and adult islet tissues, we compared the 4065 eGenes in iPSC-PPC and the 4211 eGenes in adult islets that complete summary statistics were available for. Specifically, we used the *intersect* function in R to identify eGenes that overlapped between the two tissues and *setdiff* function in R to identify eGenes that did not overlap. Similarly, using the *intersect* function in R, we compared the 22,266 expressed genes in adult islet tissues with the 4065 eGenes in iPSC-PPC to identify the proportion of iPSC-PPC eGenes that were expressed in adult islets, and vice versa with the 17,098 expressed genes in iPSC-PPC and 4211 eGenes in adult islets. The 22,266 expressed genes in adult islet tissues were obtained from the complete summary statistics uploaded by the previous study in https://zenodo.org/record/3408356.

**Comparing eQTLs present in fetal-like iPSC-PPC and adult pancreatic tissues**
Colocalization between iPSC-PPC and adult eQTLs. To identify eQTLs whose effects were driven by the same causal signals in iPSC-PPC and adult pancreatic tissues (islets and whole pancreas), we performed Bayesian colocalization using the *coloc.abf* function in *coloc* (version 5.1.0, R)[45]. Specifically, for each iPSC-PPC and adult eQTL, we tested its overlap with nearby eQTLs within 3 Mb from the gene body coordinates. eQTLs with no overlapping variants would automatically not be tested. Then, we filtered the results by requiring that each colocalization used the number of overlapping variants (called "nsnps" in the *coloc.abf* output) ≥ 500. As described above, we considered two eQTL signals to be shared if PP.H4 ≥ 80% or distinct if PP.H3 ≥ 80%. eQTL associations with PP.H4 < 80% and PP.H3 < 80% were due to insufficient power in one or both eQTL signals.

Because we, and others, have shown that $e_g$QTLs are functionally different from $e_{AS}$QTLs ($e_i$QTLs, exon eQTLs, and splicing eQTLs), we performed colocalization for $e_g$QTLs and $e_{AS}$QTLs independently (i.e., colocalization of $e_g$QTL was performed only with another $e_g$QTL and an $e_{AS}$QTL only with another $e_{AS}$QTL). All associations with PP ≥ 80% for any model are reported in Supplementary Data 9.

**Fine-mapping of adult eQTL associations.** Similarly for iPSC-PPC eQTLs, we identified candidate causal variants using the *finemap.abf* function in *coloc* (version 5.1.0, R). This Bayesian method converts *p*-values of all variants tested for a specific gene to a PP value for being the causal variant.

For all downstream analyses beyond this point, we used only iPSC-PPC, adult islets, and adult whole pancreas eQTLs that had at least one

candidate causal variant with PP ≥ 1%, were outside of the MHC region, and were annotated in GENCODE version 34 hg19, to ensure that our analyses were sufficiently powered and the multiple datasets were comparable.

**Identifying tissue-unique singleton eQTLs.** To identify tissue-unique singleton eQTLs, we obtained all eQTLs that did not colocalize with another eQTL and examined their LD with all other eQTLs of the same phenotype ($e_g$QTLs or $e_{AS}$QTLs) using their most likely candidate causal variants based on the highest PP from fine-mapping (®*nemap.abf*). If the candidate causal variant was not genotyped in the 1000 Genomes Phase 3 panel, then we used the next top candidate causal variant. We repeated this process until we found a variant that was in the 1000 Genomes or no more variants remained with causal PP 1%. Because complete summary statistics were not available for the adult conditional eQTLs, we used their lead variants publicly available from their respective studies to account for the presence of multiple causal variants in the genomic region. LD was calculated using *plink --r2 square --keep-allele-order --make-bed*[107] and the 1000 Genomes Phase 3 panel[108]. We considered two eQTLs to be in LD if their candidate causal variants were within 500 Kb and had $r^2$ 0.2. If LD could not be measured, because one of the variants was not genotyped in the 1000 Genomes, then we used distance as a metric for LD, where if the variants were within 500 Kb of each other, we considered them to be in LD. Singleton eQTLs that were found to be in LD with another eQTL (regardless of tissue) were re-annotated as "ambiguous" and excluded from downstream analyses. Otherwise, we kept their annotations as tissue-unique singletons. All annotations for singleton eQTLs are reported in Supplementary Data 10.

**Identifying eQTL modules.** eQTL modules were identified by first creating a network using the *graph_from_data_frame* function in igraph (version 1.3.4, R)[117] where the input was a data frame containing all pairs of colocalized eQTLs (nsnps ≥ 500 and PP.H4 ≥ 80%) as binary edges. We created networks for each chromosome and phenotype (gene expression and alternatively splicing) independently, totaling to 44 networks (22 chromosomes x 2 phenotypes = 44 networks). Then, we performed community detection analysis using the *cluster_leiden* function with *--objective function = "modularity", n_iterations = 500, resolution = 0.3* to identify modules of eQTLs. Upon examining them in depth, we observed that 5% of the modules contained at least one H3 association (PP.H3 ≥ 80%) between a pair of eQTLs, indicating that signals within a module were predicted to have distinct genetic variants despite being assigned to the same module. Therefore, to filter for modules that contained eQTLs likely to share the same causal variants, we required that at least 30% of all eQTL pairs had a H4 association and that the number of H4 "edges" was twice the number of H3 "edges" (number of H4 edges/ number of H3 edges ≥ 2). For example, a module with four eQTLs would have six possible pairwise combinations, and to be considered a validated module, we required at least two H4 edges and no more than one H3 edge. Modules that did not pass these thresholds were annotated as "module_failed" and excluded from downstream analyses. Summary of eQTL modules and their individual eQTL associations are reported in Supplementary Data 11. Module IDs were assigned such that the first term indicates the phenotype the module was associated with ("GE" for gene expression or "AS" for alternative splicing), the second term indicates the chromosome number, and the third term indicates a unique integer. For example, "GE_1_32" indicates that this module is associated with changes in gene expression, located in in chromosome 1, and assigned the number 32.

**Identifying tissue-unique and tissue-sharing eQTL modules.** Combinatorial eQTLs were defined in this study as an eQTL having at least one H4 association (PP.H4 ≥ 80%) with another eQTL either in the same or different tissue. These combinatorial eQTLs were then connected to

form a module, which we identified using the network analysis described above. We then categorized each module based on the activity of eQTLs in the three pancreatic tissues, having a total of seven module categories (Fig. 3b):

1. iPSC-PPC-unique: contains eQTLs in **only** iPSC-PPC
2. Adult islet-unique: contains eQTLs in **only** adult islets
3. Adult whole pancreas-unique: contains eQTLs in **only** adult whole pancreas
4. Adult-shared: contains eQTLs in adult islets **and** adult whole pancreas
5. Fetal-islet: contains eQTLs in iPSC-PPC **and** adult islets
6. Fetal-whole-pancreas: contains eQTLs in iPSC-PPC **and** adult whole pancreas
7. Fetal-adult: contains eQTLs in **all** three pancreatic tissues

We next examined the eQTLs modules for LD with eQTLs in other tissues to confirm tissue specificity. Similar to the analysis described above for identifying tissue-unique singletons, we calculated LD using *plink --r2 square --keep-allele-order --make-bed*[107] and the 1000 Genomes Phase 3 panel[108] between the eQTLs' most likely candidate causal variants (based on the highest PP; PP ≥ 1%). We considered two eQTLs to be in LD if they had an $r^2$ ≥ 0.2 and were within 500 Kb of each other. If LD could not be calculated because candidate causal variants were not genotyped in the 1000 Genomes Phase 3 panel, then we used distance as a metric for LD and considered two eQTLs to be in LD if their candidate causal variants were within 500 Kb. To account for the presence of multiple causal variants in the genomic region, we included the lead variants from the adult islet and whole pancreas conditional eQTLs in the LD comparisons to prevent misclassification of tissue-unique eQTLs.

For each of the module categories, we required that the following were true to be considered for downstream analyses:

1. iPSC-PPC-unique: contains eQTLs in only iPSC-PPC, and **all** eQTLs were not in LD with eQTLs in adult islets **and** adult whole pancreas
2. Adult islet-unique: contains eQTLs in only adult islets, and **all** eQTLs were not in LD with eQTLs in adult whole pancreas **and** iPSC-PPC
3. Adult whole pancreas-unique: contains eQTLs in only adult whole pancreas, and **all** eQTLs were not in LD with eQTLs in adult islets **and** iPSC-PPC
4. Adult-shared: contains eQTLs in only adult islets and adult whole pancreas, and **all** eQTLs were not in LD with eQTLs in iPSC-PPC
5. Fetal-islet: contains eQTLs in iPSC-PPC and adult islets, and **all** eQTLs were not in LD with eQTLs in adult whole pancreas
6. Fetal-whole-pancreas: contains eQTLs in iPSC-PPC and adult whole pancreas, and **all** eQTLs were not in LD with eQTLs in adult islets
7. Fetal-adult: contains eQTLs in **all** three pancreatic tissues.

For any module that did not meet the above requirements, we annotated the eQTLs in the module "ambiguous" and excluded for downstream analysis. Hereafter, we refer the eQTL associations in tissue-unique modules (categories 1-3) as tissue-unique combinatorial eQTLs and those in categories 5-7 as eQTLs shared between both fetal-like and adult stages. All annotations for eQTL modules and their individual eQTLs are reported in Supplementary Data 10 and Supplementary Data 11.

**Enrichment of fetal-like iPSC-PPC-unique singleton and combinatorial eQTLs in chromatin states**

We obtained chromatin state maps for human embryonic stem cell-derived pancreatic progenitor cells and adult islets from previously published studies[14,54]. Because $e_g$QTLs were likely to affect non-coding regulatory elements (Fig. 1e), we only considered them in this analysis and excluded $e_{AS}$QTLs. Enrichments for iPSC-PPC-unique singleton

and combinatorial $e_g$QTLs were calculated using a Fisher's Exact Test by comparing the proportion of fine-mapped variants (from *finemap.abf*) of the $e_g$QTLs at different thresholds of PP from 0-0.8 at 0.1 intervals to a background set of 20,000 randomly selected variants. Enrichments were Benjamini-Hochberg-corrected. Corrected *p*-values 0.05 were considered significant. Enrichment results are available in Supplementary Data 12, Fig. 3c.

## Overlap of eGenes in shared modules between fetal-like iPSC-PPC and adult pancreatic tissues

For the modules shared between both fetal-like and the two adult pancreatic tissues (categories 5-7; described above), we compared the eGenes associated with: 1) iPSC-PPC $e_g$QTLs versus adult islet $e_g$QTLs; and 2) iPSC-PPC $e_g$QTLs versus adult whole pancreas $e_g$QTLs. For $e_{AS}$QTLs, we compared the genes mapping to: 1) each isoform in iPSC-PPC versus exon in adult islets; and 2) each isoform in iPSC-PPC versus splice interval in adult whole pancreas. From these comparisons, we assigned each module an "islet_egene_overlap" label and an "whole_pancreas_egene_overlap" label in Supplementary Data 11 (also shown in Fig. 4a and Supplementary Fig. 11C), where "zero" indicates that the module did not contain an eQTL in the adult tissue, "same" indicates that the module contained eQTLs corresponding to only the same eGenes in iPSC-PPC and adult, "partial" indicates that the module contained eQTLs corresponding with partially overlapping eGenes between iPSC-PPC and adult, and "different" indicates that the module contained eQTLs corresponding to only different genes. For example, if a module was annotated with "zero" for islet_egene_overlap and "same" for whole_pancreas_egene_overlap, this meant that the module did not contain an eQTL from adult islet and had only eQTLs associated with the same eGenes between iPSC-PPC and adult whole pancreas. These annotations also meant that this module was in the "fetal-whole-pancreas" category (i.e, only contained eQTLs from iPSC-PPC and adult whole pancreas).

## Complex trait GWAS associations

**Colocalization of eQTLs with GWAS associations.** We obtained GWAS summary statistics from ten different studies: 1) type 1 diabetes[3], 2) type 2 diabetes[4], 3) body mass index[56], 4) triglycerides[56], 5) HDL cholesterol[56], 6) LDL direct[56], 7) cholesterol[56], 8) glycated hemoglobin A1C (HbA1c) levels from the MAGIC Consortium[2], 9) HbA1c levels from the Pan-UKBB Study[56], and 10) fasting glucose[2]. All of the data, except for type 1 diabetes, were provided in hg19 coordinates, therefore we converted the coordinates from hg38 to hg19 using the liftOver package in R[118]. We sorted and indexed each file using *tabix*[99]. For each trait, we performed colocalization between GWAS variants and all filtered significant eQTLs (see bolded section above) in the three pancreatic tissues with the *coloc.abf* function in *coloc* (version 5.1.0, R)[45] using *p*-values, MAF, and sample size as inputs. Then, we filtered results based on whether the lead candidate causal variant underlying both GWAS and eQTL association (from *coloc.abf* output) is genome-wide significant for GWAS association (*p*-value $\leq 5 \times 10^{-8}$) and the number of overlapping variants used to test for colocalization (nsnps) $\geq 500$. eQTLs were considered to share a genetic signal with GWAS if PP.H4 $\geq 80\%$ or have distinct signals with GWAS if PP.H3 $\geq 80\%$. For eQTL modules, we required that at least 30% of the eQTLs in the module colocalized with GWAS (PP.H4 $\geq 80\%$) and that the number of H4 associations is twice the number of H3 associations (number of H4 associations / number of H3 associations $\geq 2$). Colocalization results for the 312 GWAS loci (183 singleton and 129 module) with PP.H4 $\geq 80\%$ are available in Supplementary Data 13.

**GWAS 99% credible sets.** For each GWAS locus (based on GWAS locus ID in Supplementary Data 13), we constructed 99% credible sets with the predicted candidate causal variants underlying both eQTL and GWAS associations (from *coloc.abf* output). If the GWAS locus colocalized with a singleton eQTL, the credible sets were constructed using the output of the eQTL's colocalization with GWAS. If the GWAS locus colocalized with an eQTL module, we constructed credible sets for each of the pairwise eQTL-GWAS colocalization and retained the eQTL that resulted in the least number of candidate causal variants. If multiple eQTLs had the same number of variants in their credible set, we considered the eQTL with the highest PP.H4 for GWAS colocalization. 99% credible sets were constructed by first sorting the variants by descending order of causal PP and obtaining the least number of variants that resulted in a cumulative PP $\geq 99\%$. 99% credible sets for each of the 312 GWAS loci (183 singleton and 129 module) are reported in Supplementary Data 14.

**LD with non-pancreatic GTEx tissues.** We downloaded the lead SNPs for all significant $e_g$QTLs (including their conditionals) for the 48 non-pancreatic tissues in the GTEx dataset version 8[10] and converted their genomic positions to hg19 using UCSC liftOver[118]. We then calculated their LD with the lead SNPs of the 16 iPSC-PPC-unique $e_g$QTLs that colocalized with GWAS (Supplementary Data 15) using *plink*[107] *--tag-kb 500 --tag-r2 0.2 --show-tags all* and the 1000 Genomes[108] as the reference panel. We considered two eQTLs to be in LD if their lead SNPs were within 500 Kb and had $r^2 > 0.2$. If LD could not be calculated because the SNP was not genotyped in the reference panel, we used distance as a metric in which we considered two eQTLs to be in LD if their lead SNPs were within 500 Kb.

## Reporting summary

Further information on research design is available in the Nature Portfolio Reporting Summary linked to this article.

# Data availability

The iPSC-PPC scRNA-seq and bulk RNA-seq data generated in this study have been deposited in the GEO database under accession codes GSE152610 and GSE182758, respectively. The WGS data used in this study for iPSCORE individuals were obtained as a VCF file from phs001325.v3 [https://www.ncbi.nlm.nih.gov/projects/gap/cgi-bin/study.cgi?study_id=phs001325]. The reference gene annotation file for aligning bulk RNA-seq data of iPSC-PPC were obtained from GENCODE release version 34 in GRCh37 as a GTF file [https://www.gencodegenes.org/human/release_34.html]. The bulk RNA-seq data for iPSC, adult islet, and adult whole pancreas samples used in PCA and pseudotime analyses were obtained from phs000924, GSE50398, and phs000424, respectively. eQTL summary statistics for adult whole pancreas and islet samples were obtained from the GTEx Data Repository [https://console.cloud.google.com/storage/browser/gtex-resources] and a previously published study[11] [https://zenodo.org/record/3408356], respectively. GWAS summary statistics were obtained from the Pan UK BioBank resource [https://pan.ukbb.broadinstitute.org/], the MAGIC (Meta-Analyses of Glucose and Insulin-related traits) Consortium [https://magicinvestigators.org/downloads/; https://doi.org/10.1038/s41588-021-00852-9] the DIAMANTE Consortium [https://diagram-consortium.org/downloads.html; https://doi.org/10.1038/s41588-018-0241-6], and a previously published study[3] [http://ftp.ebi.ac.uk/pub/databases/gwas/summary_statistics/GCST90014001-GCST90015000/GCST90014023/]. Full summary statistics for iPSC-PPC eQTLs, supplemental data, and processed scRNA-seq have been deposited in Figshare [https://figshare.com/projects/Large-scale_eQTL_analysis_of_iPSC-PPC/156987].

# Code availability

Scripts for processing RNA-seq and scRNA-seq data and performing downstream analyses are publicly available at /https://github.com/jenniferngp/iPSC_PPC_eQTL_Project (version 1.0.0 of the release).

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

## Acknowledgements

This work was supported by the National Library Training Grant T15LM011271 (J.P.N., M.K.R.D., T.D.A.) and the National Institute of Diabetes and Digestive and Kidney Disease (NIDDK) F31DK131867 (J.P.N.), U01DK105541, DP3DK112155 and P30DK063491 (K.A.F.). Additional support was also received from the National Heart, Lung and Blood Institute (NHLBI) F31HL158198 (T.D.A.) and the Dongguk University Research Fund of 2023 (J.H.K.). We thank Drs. Maike Sanders and Kyle Gaulton for their advice on experimental design and analyses. This publication includes data generated at the UC San Diego IGM Genomics Center utilizing an Illumina NovaSeq 6000 that was purchased with funding from a National Institutes of Health SIG grant S10OD026929.

## Author contributions

K.A.F. and iPSCORE consortium members conceived the study. A.D.C., B.M.S., K.F. and iPSCORE consortium members performed the differentiations and generated molecular data. J.P.N., M.K.R.D. and H.M. performed quality check on scRNA-seq and RNA-seq samples. J.P.N., T.D.A., J.H.K. performed the computational analyses. K.A.F. and iPSCORE consortium members oversaw the study. J.P.N., M.D. and K.A.F. prepared the manuscript.

## Competing interests

The authors declare no competing interests.

## Additional information

## iPSCORE Consortium

Lana Ribeiro Aguiar[5], Angelo D. Arias[4], Timothy D. Arthur[2,3], Paola Benaglio[4], W. Travis Berggren[7], Juan Carlos Izpisua Belmonte[8], Victor Borja[5], Megan Cook[5], Matteo D'Antonio[2,4,5], Agnieszka D'Antonio-Chronowska[4], Christopher DeBoever[1], Kenneth E. Diffenderfer[7], Margaret K. R. Donovan[1,2], KathyJean Farnam[5], Kelly A. Frazer[4,5] ✉, Kyohei Fujita[4], Melvin Garcia[5], Olivier Harismendy[2], Benjamin A. Henson[5], David Jakubosky[2,3], Kristen Jepsen[5], He Li[4], Hiroko Matsui[5], Naoki Nariai[4], Jennifer P. Nguyen[1,2], Daniel T. O'Connor[9], Jonathan Okubo[5], Athanasia D. Panopoulos[8], Fengwen Rao[9], Joaquin Reyna[5], Bianca M. Salgado[5], Nayara Silva[4], Erin N. Smith[4], Josh Sohmer[5], Shawn Yost[1] & William. W. Young Greenwald[1]

[7]Stem Cell Core, Salk Institute for Biological Studies, La Jolla, CA 92037, USA. [8]Gene Expression Laboratory, Salk Institute for Biological Studies, La Jolla, CA 92037, USA. [9]Department of Medicine, University of California, San Diego, La Jolla, CA 92093, USA.

