## [Peer Review File · Nature Communications]

REVIEWER COMMENTS

Reviewer #1 (Remarks to the Author):

Summary and overall impression

This study explores genetic control of gene expression in an iPSC model of pancreas cell types. The authors differentiated induced pluripotent stem cells from 106 individuals into pancreatic progenitor cells (iPSC-PPC). The authors confirmed efficient differentiation into the pancreatic progenitor state using flow cytometry for the canonical markers, PDX1+ and NKX6-1+, and then performed bulk RNA-sequencing on each line. Transcriptional profiles were paired with genome-wide genotyping (by whole genome sequencing) to generate expression quantitative trait loci (eQTL) maps for the iPSC-PPCs. They then integrated these iPSC-PPC eQTL maps with published eQTL data from adult human islets (InsPIRE) and whole pancreas (GTEx) and use formal colocalization methods to identify shared and distinct eQTL signals within and between the three tissue types. Finally, they identify shared and distinct eQTL signals colocalizing with pancreas-related complex traits, including type 1 and type 2 diabetes and quantitative metabolic phenotypes.

This study provides a novel eQTL resource (the first iPSC-PPC eQTL map to our knowledge) and uses formal statistical analysis (coloc) to explore plasticity of regulatory control across human pancreas cell types and to nominate regulatory mechanisms underlying pancreas disease phenotypes. This could be an informative and useful resource. However, we have some major concerns about the colocalization method used, as well as, the interpretation and presentation of results.

Major issues

1. The results rely heavily on formal colocalization testing using the R package coloc (version 5.1.0). While this package provides a rigorous statistical approach (“coloc.susie”) for evaluating the shared causal variant hypothesis between association signals, the authors have used an older method within this package (“coloc.abf”), which assumes a single causal variant in each region for each trait. The more rigorous “coloc.susie” method relaxes this assumption, which is a critical detail, since we know that many regions harbor multiple causal variants shaping disease risk (the high prevalence of allelic heterogeneity is well-documented for complex traits, including T1D and T2D) and gene expression (e.g., GTEx and other eQTL studies have demonstrated that many eGenes have multiple independent eQTLs). Assuming a single causal variant in a region with multiple independent signals can lead to false conclusions regarding the presence or absence of colocalization with signals for another trait.
2. The authors refer to iPSC-PPC as “fetal pancreas.” However, the data shown to support these annotations (Figure S9) do not directly demonstrate that this is the case. To support this claim, we

encourage the authors compare the iPSC-PPC, human islet, and whole pancreas transcriptional profiles directly to fetal pancreas (e.g., de la O Sean et al. 2022 <https://doi.org/10.1101/2022.02.17.480942>).

3. The authors refer to human islets as “adult endocrine” and whole pancreas as “adult exocrine” tissue. While endocrine cell types tend to be abundant in human islet samples, single cell studies have demonstrated that there can be substantial representation of ductal and acinar cells in isolated human islet samples as well. Likewise, while whole pancreas is dominated by exocrine cell types, endocrine, endothelial, and immune cell types will also be partially represented in transcriptional profiles from whole pancreas. Presenting these heterogeneous tissues as “endocrine” and “exocrine” diminishes the rigor of the work. We suggest referring to the tissues that were studied (“iPSC-PPC”, “adult human islets”, and “adult whole pancreas”) when presenting the results, and providing a discussion of how these may be models of fetal, adult endocrine, and adult exocrine tissues, respectively, in the discussion section.

Minor issues

1. The readability of Figure 4a could be improved. We encourage the authors to consider using an “UpSet” plot. We also refer the authors to the GTEx V8 results (Supplement Section 14 - Tissue sharing) for inspiration on ways to visualize and quantify sharing of QTLs across tissues.

2. Typo on Line 736: 1% are available in Table S7. The eQTLs not present in this table do not having any variants with $PP \geq 1\%$

3. Typo on Line 124: eiQTLs and that their effects were commonly drive by the same causal variants while a fraction were driven by different

4. Locus plots (e.g., Fig 2C, Fig 3E-I, Fig 4B-D) should show the strength of linkage disequilibrium with index variant for all variants in the region (as is shown in Fig 6A-C).

Reviewer #2 (Remarks to the Author):

This manuscript describes a study of the impact of genetic variants on gene expression and alternative splicing in fetal stage pancreatic progenitor cells. This study is an important complement to previous studies of expression quantitative trait loci (eQTLs) in adult stage pancreatic endocrine and exocrine tissues. The study leveraged the resources in the iPSCORE consortium to differentiate iPSCs from 106 genome-sequenced individuals into pancreatic progenitor cells (PPCs). These PPCs derived from iPSCs have properties of fetal stage PPCs, and thus serve as useful model for fetal PPCs.

The need for fetal stage eQTLs in addition to those from adult stages is well justified. The methodologies for generating the RNA-seq data and determining QTLs for both genes, termed e(g)QTLs [For this review, I am enclosing subscripts in parentheses], and for splicing variant isoforms, termed e(AS)QTLs, were appropriate and performed thoroughly. The resulting resources will be of high value to the community.

The manuscript can be improved by addressing the following issues.

(1) The manuscript is long and somewhat difficult to read. Part of the latter problem results from not explaining terms when first used (see subsequent points). A large issue is the style of presentation, which is highly descriptive and detailed. The several sections in the Results are of interest, but the text in each section is very detailed, often consisting of lists of numbers of elements that are already illustrated in the Figures. A concise description of the major results in each section would be more effective.

(2) L116-122: In order to understand this section, the reader needs to know what is meant by H3 and H4 associations. Indeed, those terms refer to some hypotheses that were tested for colocalization. However, those hypotheses are not stated or explained until page 25, lines 762-767. This information needs to be explained in the Results when the H3 and H4 terminology is first used.

(3) The term or abbreviation "PP" is used extensively in the manuscript, but not until page 24 (L735) is it defined as posterior probability. This definition needs to be stated at the first usage, along with the explanation for what it is measuring (currently lines 734-735).

(4) The manuscript contrasts singleton eQTLs, apparently impacting only one gene, versus combinatorial eQTLs, impacting multiple genes, in several sections. However, the biological significance of this distinction is not explored very much. Is there some fundamental difference in how one should think about the roles of singleton versus combinatorial, other than the number of inferred target genes?

(5) L133-135: Fig. 2A does not show only 1501 eGenes overlapping between fetal-like and adult endocrine tissues. It does illustrate the extensive expression at both fetal and adult stages.

(6) What is the difference between e(i)QTLs (L93 and 100) and e(AS)QTLs (L164). Both appear to be genetic variants that affect the production of splicing isoforms (i), i.e alternative splicing (AS).

(7) The term e(g)QTL is introduced twice, initially on L93 and L99, and then later on L163. Does it mean the same thing or something different in the two places.

(8) The inference of "functional characterization" in L211-225 can work for some broad, general conclusions, but more specific analyses and comparisons are compromised by the confounding factor that three different integrative segmentation and annotations were used for the three different tissue sources. A different set of epigenetic states are assigned to each tissue, which makes comparisons difficult and impossible in some cases. This confounded situation and other factors lead to some incorrect or confusing claims. Examples include the following.

(a) L215-217: The observation that "Adult endocrine-unique and exocrine-unique combinatorials were also enriched in active chromatin states but had stronger preferences for enhancers" is supported for endocrine-unique combinatorial e(g)QTLs (Figure 3C, right panel), but the exocrine-unique combinatorial e(g)QTLs appear to be more enriched in both the active TSS and enhancer states (actually more in active TSS), according to Figure S11C.

(b) L218: It is unexpected to observe that "iPSC-PPC-unique combinatorial egQTLs were enriched in quiescent" regions in adult endocrine pancreas. The quiescent states for other tissues and cell types were depleted for many e(g)QTLs, as expected. Could the unusual enrichment in quiescent state for endocrine tissues reflect a difference in the integrative segmentation method and the decisions made for labeling the states? Note that in that latter segmentation, several other states associated with inactivity are also called, but those state were not called in other tissues.

(c) L224-2255: The conclusion about "combinatorial e(g)QTLs having the strongest preference for enhancers as observed in adult" could again be a consequence of a different segmentation, with more states defined, than the one used for the iPSC-PPC.

The best solution to these issues in point 8 is to model and assign states uniformly based on the same types of epigenetic data from all three sources. Short of doing this, it would be prudent to only draw general conclusions.

(9) L338: What are FG levels? Is it fasting glucose? Abbreviations should be explained.

(10) Figure 6: What does "PPA" refer to in the bottom panels?

Reviewer #3 (Remarks to the Author):

The study by Nguyen et al. presents a novel eQTL map in human iPSC-derived pancreatic progenitor cells (iPSC-PPC). They differentiated 106 iPSC lines previously generated and characterised by the IPSCORE consortium to iPSC-PPCs and profiled them using bulk RNA-sequencing. Expression QTLs were mapped on the level of genes, transcripts and splicing and compared to QTL data from iPSCs, pancreatic islets, and whole pancreatic tissue with the aim of characterising regulatory variation in fetal-like and adult

pancreatic tissue. Overall, the study is well-conducted, uses state-of-the-art QTL analysis methodology, and adds to the global resource of large-scale studies of iPSC-derived cell types. There is nothing particularly novel about the results (i.e. they are largely as expected in a QTL study), but nevertheless, they are a valuable resource for the community of researchers focusing on studying and modelling the genetic determinants of pancreatic diseases, such as Type 2 Diabetes. The most novel aspect of the study is the cell type (i.e. iPSC-PPC) in which the QTL mapping was conducted and the comparison of the fetal-like iPSC-PPC to adult pancreatic tissues. This comparison is likely to be of interest also to the iPSC-based disease modelling field. The manuscript is clearly written and data and results have been made available. However, while analyses are described in appropriate detail, it appears that no code was released as part of the manuscript.

Comments to the authors:

- Given the frequent cellular heterogeneity observed in iPSC differentiation, I was glad to see the authors tackling this in two ways, FACS sorting for markers of early/late iPSC-PPCs and single-cell RNA-sequencing of a subset of the differentiated lines. The authors note that in the eQTL analysis, PEER factors 1 and 4 tag the proportion of early/late iPSC-PPCs in the samples. However, since the single-cell data revealed 8 clusters/cell types, it would have been interesting to see how these cell type proportions vary across the 107 samples, e.g. via using one of the many bulk-RNA-seq deconvolution methods, and further, if any of the additional cell types present in the samples contribute to the discovered QTLs.

- In this study, the authors compare the QTLs they discovered in iPSC-PPCs to those from iPSCs and adult pancreatic tissues and define a set of QTLs that are 'tissue-unique' in the context of this particular set of tissues. They report 2683 eQTLs that are unique to the fetal-like iPSC-PPCs and 1139 eQTLs that exhibit regulatory plasticity across fetal-like and adult pancreas. As the authors note, there are issues around statistical power in this comparison, but it would nevertheless be useful to explore whether the QTLs unique to iPSC-PPCs are found in any other adult or iPSC-derived tissue (e.g. from GTEx) to confirm whether they are tissue-specific effects more generally.

- Regarding the colocalization analysis and 'resolving GWAS loci' with iPSC-PPC eQTL data: It would be good to place this analysis in some kind of context. For example, what fraction of the GWAS loci in question could be resolved with adult pancreatic data? GTEx data? iPSC-derived data? How much does the availability of iPSC-PPC add to this? Of particular interest would be those loci, that co-localise with iPSC-PPC QTLs but not with known QTLs from any other tissue.

- Was there a rationale for performing all of the analyses on the hg19 genome build? This is now a somewhat outdated genome build.

**REVIEWER COMMENTS**

**Reviewer #1 (Remarks to the Author):**

Summary and overall impression

This study explores genetic control of gene expression in an iPSC model of pancreas cell types. The authors
differentiated induced pluripotent stem cells from 106 individuals into pancreatic progenitor cells (iPSC-PPC).
The authors confirmed efficient differentiation into the pancreatic progenitor state using flow cytometry for the
canonical markers, PDX1+ and NKX6-1+, and then performed bulk RNA-sequencing on each line.
Transcriptional profiles were paired with genome-wide genotyping (by whole genome sequencing) to generate
expression quantitative trait loci (eQTL) maps for the iPSC-PPCs. They then integrated these iPSC-PPC eQTL
maps with published eQTL data from adult human islets (InsPIRE) and whole pancreas (GTEx) and use formal
colocalization methods to identify shared and distinct eQTL signals within and between the three tissue types.
Finally, they identify shared and distinct eQTL signals colocalizing with pancreas-related complex traits,
including type 1 and type 2 diabetes and quantitative metabolic phenotypes.

This study provides a novel eQTL resource (the first iPSC-PPC eQTL map to our knowledge) and uses formal
statistical analysis (coloc) to explore plasticity of regulatory control across human pancreas cell types and to
nominate regulatory mechanisms underlying pancreas disease phenotypes. This could be an informative and
useful resource. However, we have some major concerns about the colocalization method used, as well as, the
interpretation and presentation of results.

*We thank the reviewer for recognizing that our work provides a novel eQTL resource for examining human
pancreas cell types and hope that the modifications to the text we detail below satisfies their concerns.*

Major issues

1. The results rely heavily on formal colocalization testing using the R package coloc (version 5.1.0). While this
package provides a rigorous statistical approach (“coloc.susie”) for evaluating the shared causal variant hypothesis
between association signals, the authors have used an older method within this package (“coloc.abf”), which
assumes a single causal variant in each region for each trait. The more rigorous “coloc.susie” method relaxes this
assumption, which is a critical detail, since we know that many regions harbor multiple causal variants shaping
disease risk (the high prevalence of allelic heterogeneity is well-documented for complex traits, including T1D
and T2D) and gene expression (e.g., GTEx and other eQTL studies have demonstrated that many eGenes have
multiple independent eQTLs). Assuming a single causal variant in a region with multiple independent signals can
lead to false conclusions regarding the presence or absence of colocalization with signals for another trait.

*The reviewer brings up an excellent point. We agree that it is important to consider the presence of multiple causal
variants when colocalizing eQTL and GWAS signals. We had performed conditional analysis on the iPSC-PPC
eQTLs in each region to identify additional independent signals, but for adult eQTLs, the summary statistics for
the conditionals were not available, and thus we were not able to consider them in our colocalization analysis.
Without considering the adult conditionals, we agree with the reviewer that we may be misclassifying some
eQTLs that are shared with the adult tissues as unique to the iPSC-PPC.*

To resolve this concern, the reviewer recommended to use *coloc.susie*¹ which relaxes the single causal variant
assumption. We spent an extended amount of time implementing this function but was unable to run it
successfully. Fine-mapping failed for 60% of iPSC-PPC eQTLs so a full comprehensive comparison analysis
between fetal and adult pancreatic eQTLs could not be done. After several discussions with others that have
experience using *coloc.susie*, the high failure rate of fine-mapping iPSC-PPC eQTLs compared with the adult
eQTLs is likely due to the fact that their p-values are lower, due to a smaller sample size and also because we
needed to use a linear mixed model to identify eQTLs in order to account for sample relatedness. The previous
studies had used fastQTL to identify the adult eQTLs, which is more powered and hence *coloc.susie* was able to
easily detect their eQTL signals. We tried adjusting the coverage parameter in *coloc.susie* which recovered some
but not most of the iPSC-PPC eQTL signals. Of note, we found that different coverage values yielded slightly
different credible sets, and this may in turn produce different results for colocalization downstream.

However, we feel that the reviewer’s point is valid that conditional eQTLs in the adult tissues could overlap iPSC-
PPC eQTLs that we had classified as iPSC-PPC-unique. Therefore, to address the reviewer’s concern, we
downloaded the lead SNPs for the adult conditional eQTLs and incorporated them into the LD analysis, which
we have also optimized. If an iPSC-PPC-unique eQTL is in LD ($r^2 \geq 0.2$, 500kb upstream and downstream or
within 500 Kb if LD was not available) with at least one adult primary or conditional eQTL, then we classify the
eQTL as “ambiguous” and exclude it from downstream analyses. Likewise, if an adult islet-unique eQTL is in
LD with an adult pancreas primary or conditional eQTL, then we classify the eQTL as “ambiguous” and excluded
it from downstream analyses. Manuscripts using LD as an approach for identifying potentially overlapping eQTLs
between two datasets have recently been published ^{2,3}.

In the figure below, we show how the annotations changed after re-performing the LD analysis with adult
conditional eQTLs. Specifically, in the second column, we show the number of eQTLs that were previously
annotated as a tissue-unique or shared eQTL but are now annotated as “ambiguous”, meaning that they are in LD
with at least one adult conditional. In the second last row, there were 26 eQTLs that were previously annotated as
“ambiguous” that have now been recovered – this is because we slightly optimized the pipeline, however, the
majority of our previous annotations remained the same.

We have updated the following text to reflect the incorporation of this analysis. The underlined sentences indicate the phrases that specifically address the reviewer’s comment. In general, we observed similar trends as in the previous submission.

Lines 790-792	Section: Methods Sub-section: Downloading eQTL summary statistics for adult pancreatic tissues “We downloaded complete eQTL summary statistics for gene and exon associations for 420 adult human islets from the InSPIRE Consortium (https://zenodo.org/record/3408356)⁴, and gene and splicing associations for 305 adult whole pancreas from the GTEx Data Portal for GTEx Analysis version 8⁵ (https://console.cloud.google.com/storage/browser/gtex-resources). All GTEx SNPs were converted to hg19 using the UCSC liftOver Bioconductor package in R (https://www.bioconductor.org/help/workflows/liftOver/). Lead SNPs for conditional associations in the adult islets and whole pancreas datasets were downloaded from their respective studies (complete statistics were not readily available).”
Lines 828-841	Section: Methods Sub-section: Identifying tissue-unique singleton eQTLs

	* We heavily edited this section to improve readability. “To identify tissue-unique singleton eQTLs, we obtained all eQTLs that did not colocalize with another eQTL and examined their LD with all other eQTLs of the same phenotype (e_gQTLs or e_{AS}QTLs) using their most likely candidate causal variants based on the highest PP from fine-mapping (finemap.abf). If the candidate causal variant was not genotyped in the 1000 Genomes Phase 3 panel, then we used the next top candidate causal variant. We repeated this process until we found a variant that was in the 1000 Genomes or no more variants remained with causal PP $\geq 1\%$. Because complete summary statistics were not available for the adult conditional eQTLs, we used their lead variants publicly available from their respective studies to account for the presence of multiple causal variants in the genomic region. LD was calculated using plink --r2 square --keep-allele-order --make-bed¹⁰² and the 1000 Genomes Phase 3 panel¹⁰³. We considered two eQTLs to be in LD if their candidate causal variants were within 500 Kb and had $r^2 \geq 0.2$. If LD could not be measured, because one of the variants was not genotyped in the 1000 Genomes, then we used distance as a metric for LD, where if the variants were within 500 Kb of each other, we considered them to be in LD. Singleton eQTLs that were found to be in LD with another eQTL (regardless of tissue) were re-annotated as “ambiguous” and excluded from downstream analyses. Otherwise, we kept their annotations as tissue-unique singletons. All annotations for singleton eQTLs are reported in Table S11.”
Lines 876-881	Section: Methods Sub-section: Identifying tissue-unique and tissue-sharing eQTL modules “We considered two eQTLs to be in LD if they had an $r^2 \geq 0.2$ and were within 500 Kb of each other. If LD could not be calculated because candidate causal variants were not genotyped in the 1000 Genomes Phase 3 panel, then we used distance as a metric for LD and considered two eQTLs to be in LD if their candidate causal variants were within 500 Kb. To account for the presence of multiple causal variants in the genomic region, we included the lead variants from the adult islet and whole pancreas conditional eQTLs in the LD comparisons to prevent misclassification of tissue-unique eQTLs.”
Lines 182-208	Section: Results Sub-section: Developmental stage-unique and shared e_gQTLs *A large portion of this section has been updated in response to the reviewer’s comment, particularly the numbers, but we pasted a summary below. Lines 205-208: “Altogether, including e_{AS}QTLs, we identified 1,805 iPSC-PPC eQTLs that were unique to fetal-like iPSC-PPC, of which 1,518 (887 e_gQTLs + 631 e_{AS}QTLs) functioned as singletons and 287 (21 e_gQTLs + 266 e_{AS}QTLs) in modules; while 1,977 (1,175 e_gQTLs + 802 e_{AS}QTLs) were shared with adult pancreatic tissues, and 4,326 (2,066 e_gQTLs + 2,260 e_{AS}QTLs) failed one or more the stringent filters and were marked as ambiguous (Table S11, Table S12).”

Lines 237-265	Section: Results Sub-section: Regulatory plasticity in combinatorial e_gQTLs shared between fetal-like and adult pancreatic tissues *A large portion of this section has been updated in response to the reviewer’s comment, particularly the numbers, but we pasted summaries of our findings below. Lines 247-250: “These data show that 51.3% (505) of the modules shared between the iPSC-PPCs and adult pancreatic tissues regulated expression of the same genes, while 48.7% (478) displayed spatiotemporal regulatory plasticity.” Lines 261-265: “Altogether, we discovered 655 (478 e_gQTL + 177 e_{AS}QTL modules, categories C-E) shared eQTL loci that displayed regulatory plasticity in which the underlying regulatory variants were associated with one or more different genes and could thereby affect different biological processes (see Supplemental Note 2). These 655 shared eQTL loci comprise 1,043 iPSC-PPC, 934 adult islet, and 1,111 adult whole pancreas eQTL associations.”
Figures and Tables	We have updated the following figures and tables after incorporating the new eQTL annotations.  - Figure 3A, B, and C (Line 1030) - Figure 4A (Line 1052) - Figure 5A (Line 1067) - Table S10, S11, S12, S13, S14 (see legends in Supp. Material starting at Line 360)

2. The authors refer to iPSC-PPC as “fetal pancreas.” However, the data shown to support these annotations (Figure S9) do not directly demonstrate that this is the case. To support this claim, we encourage the authors to compare the iPSC-PPC, human islet, and whole pancreas transcriptional profiles directly to fetal pancreas (e.g., de la O Sean et al. 2022 <https://doi.org/10.1101/2022.02.17.480942>).

We understand the reviewer’s concern. The fetal pancreas data in de la O Sean et al. 2022 is not publicly available. However, we thought it was a great study and have referenced it in our manuscript (see Lines 48-53, reference #28).

To address the reviewer’s comment, we have examined the expression levels of known fetal pancreatic genes⁸⁻¹² (*PDX1*, *NKX6-1*, *SOX9*, and *DLK1*) in scRNA-seq. We found that the main PPC cluster highly expressed these markers (see panels A-D in figure below), confirming that these cells represent an earlier stage of pancreas development. These results are also consistent with a previous study that used the same genes to benchmark their iPSC-derived PPC samples that were derived using the same protocol (Rezania et al., 2018¹³) to the human fetal pancreas (Figure 3E in Goncalves et al., Nature, 2021¹⁴).

We also compared the expression levels of these genes in bulk RNA-seq of iPSC-PPC and the two adult pancreatic
tissues and show that they were expressed at higher levels in the iPSC-PPC (see **panels E-H** below). Hence our
findings are congruent with Goncalves et al., 2021¹⁴, which concluded that their iPSC-PPC samples have fetal-
like molecular properties.

Below, we summarized references that used stem cell-derived pancreatic progenitor cells to model human
pancreas development.

1) Gonçalves, Carla A., et al. "A 3D system to model human pancreas development and its reference single-
cell transcriptome atlas identify signaling pathways required for progenitor expansion." *Nature*
*communications* 12.1 (2021): 3144. (<https://doi.org/10.1038/s41467-021-23295-6>)

a) This study compared iPSC-derived pancreatic progenitors to the fetal pancreas and showed that while
the derived cells were enriched for liver markers, they also expressed pancreatic markers observed in fetal
pancreas, including *PDX1* and *SOX9*. They also compared between iPSC-derived pancreatic progenitors
grown in 2D and 3D and found that expression profiles between the two are very similar, but 3D more
closely resembles the human fetal pancreas.

2) Geusz, Ryan J., et al. "**Pancreatic progenitor epigenome maps prioritize type 2 diabetes risk genes with**
**roles in development.**" *Elife* 10 (2021): e59067. (<https://elifesciences.org/articles/59067>)¹⁵

a) This study utilized hESC-derived pancreatic progenitors to study type 2 diabetes risk variants specifically
active during pancreas development

3) Ramond, Cyrille, et al. "**Understanding human fetal pancreas development using subpopulation**
**sorting, RNA sequencing and single-cell profiling.**" *Development* 145.16 (2018): dev165480.
(<https://doi.org/10.1242/dev.165480>)¹⁶

a) This study showed that *in vitro*-produced endocrine cells (using the same protocol; Rezania et al., 2018
127¹³) followed paths similar to their *in-vivo* counterparts.

For these reasons, we believe that the iPSC-PPCs serve as a suitable model for the fetal pancreas. We have updated
the Introduction to describe these previous body of literature (see **Lines 48-53**) as well as added the panels below
into **Figure S9 (Line 181 in Supplemental Material)** and added the following sentence into the Results (**Line 89-**
**91, underlined below**).

Line 89-91	Section: Results Sub-section: Large-scale differentiation of fetal-like pancreatic progenitor cells "To examine the similarities between iPSC-PPC and adult pancreatic transcriptomes, we generated bulk RNA-seq for all 107 iPSC-PPC samples and inferred the pseudotime on each sample, along with 213 iPSCs ^{35,37} , 87 pancreatic islets ³⁸ , and 176 whole pancreatic tissues ³⁹ . Pseudotime analysis and comparative expression analysis of early developmental genes showed that the iPSC-PPC samples corresponded to an early timepoint of pancreas development (Figure S9, Table S6). "
------------	--

**Figure Legend: (a-d)** UMAP plots showing the expression of fetal pancreatic genes^{9-12,17-19} (*PDX1*, *NKX6-1*,
*SOX9*, and *DLK1*) in early and late PPCs in iPSC-PPC (cell types labeled in **Figure S4A**). **(e-h)** Box plots showing
the expression of the same fetal pancreatic genes in panels **c-f** in bulk RNA-seq of 107 iPSC-PPCs (green), 87
adult islets (blue), and 176 adult whole pancreas (yellow). Paired student T-test was performed to evaluate the
significance of expression differences between tissues. * = $p < 0.05$, ** = $p < 0.01$, *** = $p < 0.001$, **** = $p <$
0.0001 . Box plots indicate median interquartile range (IQR), and 1.5 x IQR.

3. The authors refer to human islets as “adult endocrine” and whole pancreas as “adult exocrine” tissue. While
endocrine cell types tend to be abundant in human islet samples, single cell studies have demonstrated that there
can be substantial representation of ductal and acinar cells in isolated human islet samples as well. Likewise,
while whole pancreas is dominated by exocrine cell types, endocrine, endothelial, and immune cell types will also
be partially represented in transcriptional profiles from whole pancreas. Presenting these heterogeneous tissues as
“endocrine” and “exocrine” diminishes the rigor of the work. We suggest referring to the tissues that were studied
(“iPSC-PPC”, “adult human islets”, and “adult whole pancreas”) when presenting the results, and providing a
discussion of how these may be models of fetal, adult endocrine, and adult exocrine tissues, respectively, in the
discussion section.

Thank you for this suggestion. We agree that referring to human islets as “adult endocrine” and whole pancreas
as “adult exocrine” is not entirely accurate. We have modified the manuscript to replace adult endocrine with
“adult islets” and adult exocrine with “adult whole pancreas”.

Minor issues

1. The readability of Figure 4a could be improved. We encourage the authors to consider using an “UpSet” plot.
We also refer the authors to the GTEx V8 results (Supplement Section 14 - Tissue sharing) for inspiration on
ways to visualize and quantify sharing of QTLs across tissues.

7

Thank you for this suggestion. We have updated **Figure 4A** (Line 1052) with an Upset plot (also pasted below)
and agree that this improves the readability of the figure.

2. Typo on Line 736: 1% are available in Table S7. The eQTLs not present in this table do not having any variants
with $PP \geq 1\%$

We have updated the sentence to change from “having” to “have” (now still Line 740).

3. Typo on Line 124: eiQTLs and that their effects were commonly drive by the same causal variants while a
fraction were driven by different

The typo has been fixed from “drive” to “driven”. Please see Line 128.

4. Locus plots (e.g., Fig 2C, Fig 3E-I, Fig 4B-D) should show the strength of linkage disequilibrium with index
variant for all variants in the region (as is shown in Fig 6A-C).

Thank you for this suggestion. We have updated the locus plots to show the strength of linkage disequilibrium
with the index variant.

**Reviewer #2 (Remarks to the Author):**

This manuscript describes a study of the impact of genetic variants on gene expression and alternative splicing in
fetal stage pancreatic progenitor cells. This study is an important complement to previous studies of expression

quantitative trait loci (eQTLs) in adult stage pancreatic endocrine and exocrine tissues. The study leveraged the
resources in the iPSCORE consortium to differentiate iPSCs from 106 genome-sequenced individuals into
pancreatic progenitor cells (PPCs). These PPCs derived from iPSCs have properties of fetal stage PPCs, and thus
serve as useful model for fetal PPCs.

The need for fetal stage eQTLs in addition to those from adult stages is well justified. The methodologies for
generating the RNA-seq data and determining QTLs for both genes, termed e(g)QTLs [For this review, I am
enclosing subscripts in parentheses], and for splicing variant isoforms, termed e(AS)QTLs, were appropriate and
performed thoroughly. The resulting resources will be of high value to the community.

We thank the reviewer for their encouraging comments about the value of our study.

The manuscript can be improved by addressing the following issues.

(1) The manuscript is long and somewhat difficult to read. Part of the latter problem results from not explaining
terms when first used (see subsequent points). A large issue is the style of presentation, which is highly descriptive
and detailed. The several sections in the Results are of interest, but the text in each section is very detailed, often
consisting of lists of numbers of elements that are already illustrated in the Figures. A concise description of the
major results in each section would be more effective.

We thank the reviewer for these suggestions on how to improve our manuscript. We have spent a substantial
amount of time in meticulously editing the manuscript to address the reviewer's comments, and we hope the
reviewer will find the revised manuscript easier to read. We have eliminated information from the results that
were redundant with the methods. Additionally, we have moved certain numbers from the text to the figure
legends to further improve the readability. Our manuscript contains many analyses that are difficult to follow
without being explicit, but we have tried to make the descriptions more concise.

(2) L116-122: In order to understand this section, the reader needs to know what is meant by H3 and H4
associations. Indeed, those terms refer to some hypotheses that were tested for colocalization. However, those
hypotheses are not stated or explained until page 25, lines 762-767. This information needs to be explained in the
Results when the H3 and H4 terminology is first used.

We have updated the manuscript to define the five hypothesis models from *coloc.abf* earlier in the text. Please see
below.

Lines 117-123	Section: Results Sub-section: Identification and characterization of gene and isoform eQTLs in fetal-like iPSC-PPCs “Coloc.abf”²⁰ employs a Bayesian approach to estimate the PP that each of the five colocalization models best explains the association between two genetic signals: H0) no associations detected in either signal; H1) association detected in only signal 1; H2) association detected in only signal 2; H3) associations detected in both signals but driven by different causal variants, and H4) associations detected in both signals and driven by the same causal variant. We identified 410 (40.7%) genes that had at least one H4 (PP.H4, posterior
---------------	---

	probability for H4 \geq 80%) or H3 (PP.H3, posterior probability for H3 \geq 80%) association between their e _g QTL and e _i QTLs, of which the majority (333, 81.2%) had only overlapping signals (all H4), 38 (9.3%) had only non-overlapping signals (all H3), and 39 (9.5%) had both overlapping and non-overlapping e _i QTLs (both H3 and H4; an e _g QTL can colocalize with an e _i QTL corresponding to one isoform but not with another e _i QTL corresponding to a second isoform) (Figure 1G, Table S10).”
--	---

(3) The term or abbreviation "PP" is used extensively in the manuscript, but not until page 24 (L735) is it defined as posterior probability. This definition needs to be stated at the first usage, along with the explanation for what it is measuring (currently lines 734-735).

We agree with the reviewer. We had first defined “PP” in Line 106 in the original manuscript when describing Figure 1F but did not explain what it was measuring. We have updated the text accordingly (see below).

Line 108	Section: Results Sub-section: Identification and characterization of gene and isoform eQTLs in fetal-like iPSC-PPCs “We additionally estimated the transcription factor (TF) binding score for each variant using the Genetic Variants Allelic TF Binding Database ²¹ and found that, at increasing posterior probability (PP, probability that the variant is causal for the association) thresholds, the candidate causal variants underlying e_gQTLs were more likely to affect TF binding compared to those underlying e_iQTLs (Figure 1F, Table S8, Table S9).”
----------	--

(4) The manuscript contrasts singleton eQTLs, apparently impacting only one gene, versus combinatorial eQTLs, impacting multiple genes, in several sections. However, the biological significance of this distinction is not explored very much. Is there some fundamental difference in how one should think about the roles of singleton versus combinatorial, other than the number of inferred target genes?

Singleton eQTLs are all unique spatiotemporally (tissue and temporally) while combinatorial eQTLs tend to be shared either across the fetal-like iPSC-PPCs, adult islets and adult whole pancreas tissues. Furthermore, the singleton eQTLs were more enriched for promoter regions, which is consistent with their regulation of a single gene, and the combinatorial eQTLs were more enriched in stretch enhancers, which is consistent with their regulation of multiple genes. We have tried to highlight these differences better in the revised manuscript with the updated analysis in response to this question. See Lines 210-222 (also pasted below) as well as Figure 3C-D (Line 1030, or below). Therefore, we identified some biological differences in the regulatory variants in these two eQTL classes and suspect that there could be additional differences that are yet to be discovered.

Lines 210-222	Section: Results
---------------	-------------------------

Sub-section: **Functional enrichment and characterization of fetal-like tissue-unique e_gQTLs**

“At high PP thresholds, we observed the strongest enrichment of singleton e_gQTLs in active promoter (TssA) regions, consistent with their role in regulating the expression of a single gene (**Figure 3C, Table S13**). For combinatorial e_gQTLs, we observed a strong enrichment in PPC-specific stretch enhancer (PSSE) regions at high PP thresholds ($p = 0.001$, OR = 1,345, PP threshold = 60%) (**Figure 3C, Table S13**), consistent with their involvement in the transcriptional regulation of multiple genes. We also evaluated the enrichment of fetal-like iPSC-PPC-specific singleton and combinatorial e_gQTLs in adult islet chromatin states⁴⁸ (**Figure 3C**). No meaningful enrichments were observed for fetal-like iPSC-PPC-unique combinatorial e_gQTLs, but iPSC-PPC-unique singleton e_gQTLs were enriched in adult promoter regions ($p = 4.1 \times 10^{-4}$, OR = 28.4, PP threshold = 80%). For example, we observed that the iPSC-PPC-unique singleton e_gQTL in the *CDC37L1-DT* locus overlapped an active promoter region in the adult islet, while in both adult islet and adult whole pancreas, the variants in the same region are not active (**Figure 3D**). Overall, these results showed that the e_gQTLs annotated as iPSC-PPC tissue-unique were enriched in regulatory elements consistent with their proposed functions.”

Figure Legend: (c) Top panels: Enrichment (odds ratio) of iPSC-PPC singleton and combinatorial e_gQTLs in hESC-derived PPC chromatin states²¹. Bottom panels: Enrichment (odds ratio) of iPSC-PPC singleton and combinatorial e_gQTLs in adult islet chromatin states⁴⁹. Enrichment was calculated using a two-sided Fisher’s Exact Test comparing the proportion of candidate causal variants overlapping the chromatin states versus a background of randomly selected 20,000 variants at various PP thresholds. P-values were Benjamini-Hochberg-corrected and considered significant if the corrected p-values < 0.05 (indicated by asterisk). (d) *CDC27L1-DT* locus showing an iPSC-PPC-unique singleton e_gQTL overlapping an adult islet active promoter region. Lower panel shows positions of active promoters in the adult islets.

(5) L133-135: Fig. 2A does not show only 1501 eGenes overlapping between fetal-like and adult endocrine tissues. It does illustrate the extensive expression at both fetal and adult stages.

The iPSC-PPCs and adult islets share extensive gene expression with one another. However, we observed little overlap of eGenes between the two tissues, which was not expected given the large overlap of genes expressed at both stages. This shows that the impact of genetic variants on gene expression is dependent on the developmental stage of the pancreas.

We have added the Venn Diagram (shown on the right) as **Figure 2A** to show the 1,501 eGene overlap between iPSC-PPC and adult islets. We've also updated the text with the appropriate figure reference (see Line 142).

(6) What is the difference between e(i)QTLs (L93 and 100) and e(AS)QTLs (L164).

Both appear to be genetic variants that affect the production of splicing isoforms (i), i.e alternative splicing (AS).

That is correct, e_iQTL and e_{AS}QTLs both affect alternative splicing. However, we use “e_iQTLs” to refer specifically to the isoform eQTLs we called in iPSC-PPC, and “e_{AS}QTLs” to refer to all eQTLs associated with alternative splicing, which includes e_iQTL (from iPSC-PPC), exon eQTLs (from adult islets), and splicing eQTLs (from adult pancreas). We believe it is important to distinguish e_iQTLs from those called in the adult pancreatic tissues because the adult e_{AS}QTLs (exon and splicing) were derived using different methodologies. Specifically, e_{AS}QTLs in adult islets were called using exon expression and e_{AS}QTLs in adult pancreas were called using junction read counts. We have clarified this in the following three sections (listed below):

Lines 168-171	Section: Results Sub-section: Developmental stage-unique and shared e_gQTLs “Due to the many different types of eQTLs used in this study, we refer to all eQTLs as a collective unit as “eQTLs”, eQTLs that were associated with gene expression as “e_gQTLs” (as defined above), and eQTLs associated with changes in alternative splicing (e_iQTLs, exon eQTLs, and sQTLs) as “e_{AS}QTLs”.”
Lines 793-794	Section: Methods Sub-section: Downloading eQTL summary statistics for adult pancreatic tissues “Due to the many different types of eQTLs used in this study, we refer to all eQTLs as a collective unit as “eQTLs”, eQTLs that were associated with gene expression as “e_gQTLs” (as defined above), and eQTLs associated with changes in alternative splicing (e_iQTLs, exon eQTLs, and sQTLs) as “e_{AS}QTLs”.”
Lines 814-815	Section: Methods Sub-section: Colocalization between iPSC-PPC and adult eQTLs

“Because we, and others, have shown that e_gQTLs are functionally different from e_{AS}QTLs (e_iQTLs, exon eQTLs, and splicing eQTLs), we performed colocalization for e_gQTLs and e_{AS}QTLs independently (i.e., colocalization of e_gQTL was performed only with another e_gQTL and an e_{AS}QTL only with another e_{AS}QTL).”

(7) The term e(g)QTL is introduced twice, initially on L93 and L99, and then later on L163. Does it mean the
same thing or something different in the two places.

Thank you for this question. They mean the same thing in both places. We wanted to define the terms a second
time for clarity. We revised the following sentence to be more clear (see underlined phrase below).

Line 170

Section: **Results**

Sub-section: **Developmental stage-unique and shared e_gQTLs**

“Due to the many different types of eQTLs used in this study, we refer to all eQTLs as a collective unit as “eQTLs”, eQTLs that were associated with gene expression as “e_gQTLs” (as defined above), and eQTLs associated with changes in alternative splicing (e_iQTLs, exon eQTLs, and sQTLs) as “e_{AS}QTLs”.

(8) The inference of "functional characterization" in L211-225 can work for some broad, general conclusions, but
more specific analyses and comparisons are compromised by the confounding factor that three different
integrative segmentation and annotations were used for the three different tissue sources. A different set of
epigenetic states are assigned to each tissue, which makes comparisons difficult and impossible in some cases.
This confounded situation and other factors lead to some incorrect or confusing claims. Examples include the
following.

(a)L215-217: The observation that "Adult endocrine-unique and exocrine-unique combinatorials were also
enriched in active chromatin states but had stronger preferences for enhancers" is supported for endocrine-unique
combinatorial e(g)QTLs (Figure 3C, right panel), but the exocrine-unique combinatorial e(g)QTLs appear to be
more enriched in both the active TSS and enhancer states (actually more in active TSS), according to Figure S11C.

(b) L218: It is unexpected to observed that “iPSC-PPC-unique combinatorial egQTLs were enriched in quiescent”
regions in adult endocrine pancreas. The quiescent states for other tissues and cell types were depleted for many
e(g)QTLs, as expected. Could the unusual enrichment in quiescent state for endocrine tissues reflect a difference
in the integrative segmentation method and the decisions made for labeling the states? Note that in that latter
segmenation, several other states associated with inactivity are also called, but those state were not called in other
tissues.

(c) L224-2255: The conclusion about "combinatorial e(g)QTLs having the strongest preference for enhancers as
observed in adult" could again be a consequence of a different segmentation, with more states defined, than the
one used for the iPSC-PPC.

The best solution to these issues in point 8 is to model and assign states uniformly based on the same types of epigenetic data from all three sources. Short of doing this, it would be prudent to only draw general conclusions.

We thank the reviewer for this thorough critique. We agree that it is best to only draw general conclusions given the different annotations for the three pancreatic tissues. We have redone the analysis in response to the reviewer's comment #4 (Line 231) and updated the Results section in Lines 210-222 accordingly, which is pasted below:

Lines 210-222	Section: Results Sub-section: Functional enrichment and characterization of tissue-unique e_gQTLs “At high PP thresholds, we observed the strongest enrichment of singleton e_gQTLs in active promoter (TssA) regions, consistent with their role in regulating the expression of a single gene (Figure 3C, Table S13). For combinatorial e_gQTLs, we observed a strong enrichment in PPC-specific stretch enhancer (PSSE) regions at high PP thresholds (p = 0.001, OR = 1,345, PP threshold = 60%) (Figure 3C, Table S13), consistent with their involvement in the transcriptional regulation of multiple genes. We also evaluated the enrichment of fetal-like iPSC-PPC-specific singleton and combinatorial e_gQTLs in adult islet chromatin states⁴⁸ (Figure 3C). No meaningful enrichments were observed for fetal-like iPSC-PPC-unique combinatorial e_gQTLs, but iPSC-PPC-unique singleton e_gQTLs were enriched in adult promoter regions (p = 4.1 x 10⁻⁴, OR = 28.4, PP threshold = 80%). For example, we observed that the iPSC-PPC-unique singleton e_gQTL in the CDC37L1-DT locus overlapped an active promoter region in the adult islet, while in both adult islet and adult whole pancreas, the variants in the same region are not active (Figure 3D). Overall, these results showed that the e_gQTLs annotated as iPSC-PPC tissue-unique were enriched in regulatory elements consistent with their proposed functions.”
---------------	--

(9) L338: What are FG levels? Is it fasting glucose? Abbreviations should be explained.

Thank you for pointing this out. FG levels refer to fasting glucose. We have provided a definition for the abbreviation in the text (see Lines 272).

(10) Figure 6: What does "PPA" refer to in the bottom panels?

We thank the reviewer for pointing out that we had used “PPA” in the figure panels. “PPA” means posterior probability of association. We had meant to use “PP”. We have updated Figures 6 and 7 (Line 1073, 1092) to change the labels to “PP”.

Reviewer #3 (Remarks to the Author):

The study by Nguyen et al. presents a novel eQTL map in human iPSC-derived pancreatic progenitor cells (iPSC-PPC). They differentiated 106 iPSC lines previously generated and characterised by the IPSCORE consortium to iPSC-PPCs and profiled them using bulk RNA-sequencing. Expression QTLs were mapped on the level of genes, transcripts and splicing and compared to QTL data from iPSCs, pancreatic islets, and whole pancreatic tissue with

334 the aim of characterising regulatory variation in fetal-like and adult pancreatic tissue. Overall, the study is well-
 335 conducted, uses state-of-the-art QTL analysis methodology, and adds to the global resource of large-scale studies
 of iPSC-derived cell types. There is nothing particularly novel about the results (i.e. they are largely as expected
 in a QTL study), but nevertheless, they are a valuable resource for the community of researchers focusing on
 studying and modelling the genetic determinants of pancreatic diseases, such as Type 2 Diabetes. The most novel
 aspect of the study is the cell type (i.e. iPSC-PPC) in which the QTL mapping was conducted and the comparison
 of the fetal-like iPSC-PPC to adult pancreatic tissues. This comparison is likely to of interest also to the iPSC-
 based disease modelling field. The manuscript is clearly written and data and results have been made available.
 However, while analyses are described in appropriate detail, it appears that no code was released as part of the
 manuscript.

 We thank the reviewer for recognizing the novel aspects of our study and providing highly insightful suggestions.
 We have incorporated the suggested analyses into our manuscript and believe it has greatly improved the impact
 of our paper. These additions have provided valuable insights into the spatiotemporal contexts of pancreatic
 eQTLs and as a result, we have updated the title of our manuscript (see below). Per the reviewer’s suggestion, we
 have also uploaded our code onto Github ([/https://github.com/jenniferngp/iPSC_PPC_eQTL_Project](https://github.com/jenniferngp/iPSC_PPC_eQTL_Project)) and added
 a “Code Availability” section (see Line 972, also pasted below).

Manuscript Title	“Spatiotemporally informed pancreatic eQTLs provides novel insights into obesity and diabetes etiology”
Line 972	Section: Code Availability “Scripts for processing RNA-seq and scRNA-seq data and performing downstream analyses are available at /https://github.com/jenniferngp/iPSC_PPC_eQTL_Project.”

Comments to the authors:

1. Given the frequent cellular heterogeneity observed in iPSC differentiation, I was glad to see the authors tackling
 this in two ways, FACS sorting for markers of early/late iPSC-PPCs and single-cell RNA-sequencing of a subset
 of the differentiated lines. The authors note that in the eQTL analysis, PEER factors 1 and 4 tag the proportion of
 early/late iPSC-PPCs in the samples. However, since the single-cell data revealed 8 clusters/cell types, it would
 have been interesting to see how these cell type proportions vary across the 100 samples, e.g. via using one of the
 many bulk RNA-seq deconvolution methods, and further, if any of the additional cell types present in the samples
 contribute to the discovered QTLs.

The reviewer posed an excellent question. To estimate the cell type proportions in bulk, we deconvoluted the bulk
 samples using CIBERSORTx²³. As expected, we observed that the estimated fraction of late PPCs by
 CIBERSORTx²³ strongly correlated with the percentage of late PPCs measured by FACS ($R = 0.702$, $p < 2.2 \times 10^{-16}$; see figure below). We have added the figure to our manuscript and updated the following text (see table below) to reflect the incorporation. We have also added a supplemental table containing the deconvoluted cell type fractions and the signature matrix.

We also agree with the reviewer that it would be interesting to understand how the different cell types in the iPSC-
 PPCs contribute to the QTLs. However, performing this analysis would be challenging as the gene expression
 profiles between the cell types are highly similar. Furthermore, the late PPCs comprise the majority of cells
 (>80%), therefore the cell type-specific genetic effects on gene expression may be overpowered by late PPC.
 Thus, we believe we do not have the power to detect the impact of cell type specific regulatory variants on gene
 expression.

Line 154 in Supplemental Material	Section: Figure S7B  Legend: “We next examined the correspondence between the percentage of late PPCs from FACS (PDX1+/NKX6-1+) (Y-axis; Table S2, Figure S2, Figure S3) and the estimated percentage of late PPCs (including replicating late PPCs) in bulk RNA-seq using CIBERSORTx²³ (X-axis; Table S5). As expected, we observed a strong correlation between the two measurements ($R = 0.702$, $p < 2.2 \times 10^{-16}$), indicating that the cellular heterogeneity captured by flow cytometry is represented in the iPSC-PPC bulk expression profiles.”
Line 669	Section: Methods Sub-section: Cellular deconvolution “For each of the eight cell types in scRNA-seq, we selected the top 200 most differentially expressed genes that were unique to the cell type (i.e., not expressed in the other cell types). Replicating late PPCs and late PPCs had many overlapping expressed genes so fewer (n = 16 and 164, respectively) were selected. We obtained the average expression of the signature genes for each cell type using AverageExpression in Seurat and provided it as input into CIBERSORTx²³ (https://cibersortx.stanford.edu/) along with bulk TPM matrix. Batch correction and quantile normalization were both disabled. We ran CIBERSORTx²³ deconvolution on absolute mode with at least 100 permutations. The predicted fraction of

	late PPCs and replicating late PPCs were compared to FACS measurements of double-positive PDX-1 ⁺ /NKX6-1 ⁺ cells (Figure S7B; Table S5).”
Lines 31-35 in Supplemental Material	Section: Supplemental Note 1 Sub-section: Characterization of iPSC-PPC fetal-like transcriptomes using scRNA-seq “We also deconvoluted cell type proportions in bulk RNA-seq samples of iPSC-PPC using cell type-specific markers in scRNA-seq and found that the estimated proportions for late PPC highly corresponded with flow cytometry measurements of PDX1 ⁺ /NKX6-1 ⁺ cells (R = 0.702, p < 2.2x10 ⁻¹⁶ , Pearson’s correlation, Figure S7B, Table S5). These results show that the flow cytometry analysis accurately captured the fraction of late PPCs in the 107 iPSC-PPCs.”

2. In this study, the authors compare the QTLs they discovered in iPSC-PPCs to those from iPSCs and adult
pancreatic tissues and define a set of QTLs that are ‘tissue-unique’ in the context of this particular set of tissues.
They report 2683 eQTLs that are unique to the fetal-like iPSC-PPCs and 1139 eQTLs that exhibit regulatory
plasticity across fetal-like and adult pancreas. As they authors note, there are issues around statistical power in
this comparison, but it would nevertheless be useful to explore whether the QTLs unique to iPSC-PPCs are found
in any other adult or iPSC-derived tissue (e.g. from GTEx) to confirm whether they are tissue-specific effects
more generally.

We thank the reviewer for this and the following suggestion in comment #3 (**Line 391 in this document**). We have
calculated the LD between the lead variants of the 16 iPSC-PPC-unique e_gQTLs that colocalized with GWAS
signals (see answer to comment #3 below) and the e_gQTLs in the 48 other tissues in the GTEx dataset (version 8)
⁵. We considered two e_gQTLs to be in LD if their lead variants had r² ≥ 0.2 within 500 Kb or within 500 Kb if
LD could not be calculated. We found that 8 (50% of 16) of the iPSC-PPC-unique e_gQTLs were not in LD, one
of which includes the *TPD52* e_gQTL that we later describe in the last section. We also note that the number of
iPSC-PPC-unique eQTLs have decreased since the last submission in response to Reviewer #1’s comment #1
(**Line 29 in this document**).

We have added the following to the revised manuscript:

Line 950	Section: Methods Sub-section: LD with non-pancreatic GTEx tissues “We downloaded the lead SNPs for all significant e _g QTLs (including their conditionals) for the 48 non-pancreatic tissues in the GTEx dataset version 8 ¹⁰ and converted their genomic positions to hg19 using UCSC liftOver ¹¹⁶ . We then calculated their LD with the lead SNPs of the 16 iPSC-PPC-unique e _g QTLs that colocalized with GWAS (Table S16) using plink ¹⁰² <code>--tag-kb 500 --tag-r2 0.2 --show-tags all</code> and the 1000 Genomes ¹⁰³ as the reference panel. We considered two eQTLs to be in LD if their lead SNPs were within 500 Kb and had r ² ≥ 0.2. If LD could not be calculated because the SNP was not genotyped in the reference panel, we used distance as a metric in which we considered two eQTLs to be in LD if their lead SNPs were within 500 Kb.”
----------	--

Lines 316-328	Section: Results Sub-section: Spatiotemporally informed eQTL resource provides novel insights into GWAS signals “To further assess the utility of our spatiotemporally informed eQTL resource, we next examined GWAS signals that could only be interpreted by including fetal pancreatic eQTLs. We calculated the fraction of GWAS loci that colocalized with only iPSC-PPC eQTLs, only adult islet eQTLs, and only adult whole pancreas eQTLs. For fair comparisons, we considered only e_gQTLs in this assessment. Of the 191 GWAS loci that colocalized with an e_gQTL (Figure S12, Table S14), we found that 13% (24 loci) colocalized with 16 iPSC-PPC-unique e_gQTLs, 25% (47) with 27 adult islet-unique e_gQTLs, and 28% (53) with 46 adult whole pancreas-unique e_gQTLs. The remaining 35% (67) GWAS loci colocalized with 121 e_gQTLs shared between multiple tissues. We next determined how many of the 16 iPSC-PPC-unique e_gQTLs were active in 48 non-pancreatic tissues in the GTEx study¹⁰. We calculated LD ($r^2 > 0.2$ within 500 Kb or within 500 Kb if LD information was not available) between the lead variants of each of the 16 iPSC-PPC-unique e_gQTL and each e_gQTL for the non-pancreatic tissues in GTEx. We identified 8 (31% of 16, all singletons) that were independent and exclusive to fetal-like iPSC-PPC (i.e., did not have LD; Table S16). One of these 8 iPSC-PPC e_gQTLs (TPD52 e_gQTL) is described below in further detail. These results show that integrating fetal-like iPSC-PPC eQTLs can help resolve certain GWAS loci that cannot be resolved using only adult datasets.”
Line 452 in Supplemental Material	Table S16: LD analysis with non-pancreatic GTEx tissues “This table contains results for the LD analysis conducted between the 16 iPSC-PPC-unique e_gQTLs that colocalized with GWAS signals and the e_gQTLs in the 48 non-pancreatic tissues in the GTEx dataset version 8⁵. We provide the eQTL ID (assigned as [tissue_type]_[discovery_order]_[transcript_ID]), eQTL phenotype, eQTL type (singleton or combinatorial), transcript ID, gene ID, gene name, category annotation, and a TRUE/FALSE indicating whether the e_gQTL is in LD with an adult e_gQTL.”

3. Regarding the colocalization analysis and ‘resolving GWAS loci’ with iPSC-PPC eQTL data: It would be good to place this analysis in some kind of context. For example, what fraction of the GWAS loci in question could be resolved with adult pancreatic data? GTEx data? iPSC-PPC derived data? How much does the availability of iPSC-PPC add to this? Of particular interest would be those loci, that co-localise with iPSC-PPC QTLs but not with known QTLs from any other tissue.

We agree with the reviewer that it is important to evaluate the utility of iPSC-PPC eQTL data, as well as data generated from adult islets and adult whole pancreas. In response, we focused on the 191 GWAS loci that colocalized with an e_gQTL and assessed the fraction that could be resolved with using only iPSC-PPC derived data, only adult islet data, or only adult whole pancreas data. We found that 13% (24 loci) of the GWAS loci colocalized with 16 iPSC-PPC-unique e_gQTLs while 25% (47) colocalized with 27 adult islet-unique e_gQTLs and 28% (53) with 46 adult pancreas-unique e_gQTLs. The remaining 35% (67) GWAS loci colocalized with 121 e_gQTL signals shared between multiple tissues. This suggests that there are certain GWAS loci that was resolved

by using iPSC-PPC data that cannot be resolved using the adult datasets. We believe this finding is very insightful
and therefore have incorporated it in the manuscript (see below).

Lines 316-328	Section: Results Sub-section: Spatiotemporally informed eQTL resource provides novel insights into GWAS signals “To further assess the utility of our spatiotemporally informed eQTL resource, we next examined GWAS signals that could only be interpreted by including fetal pancreatic eQTLs. We calculated the fraction of GWAS loci that colocalized with only iPSC-PPC eQTLs, only adult islet eQTLs, and only adult whole pancreas eQTLs. For fair comparisons, we considered only e_gQTLs in this assessment. Of the 191 GWAS loci that colocalized with an e_gQTL (Figure S12, Table S14), we found that 13% (24 loci) colocalized with 16 iPSC-PPC-unique e_gQTLs, 25% (47) with 27 adult islet-unique e_gQTLs, and 28% (53) with 46 adult whole pancreas-unique e_gQTLs. The remaining 35% (67) GWAS loci colocalized with 121 e_gQTLs shared between multiple tissues. We next determined how many of the 16 iPSC-PPC-unique e_gQTLs were active in 48 non-pancreatic tissues in the GTEx study¹⁰. We calculated LD ($r^2 > 0.2$ within 500 Kb or within 500 Kb if LD information was not available) between the lead variants of each of the 16 iPSC-PPC-unique e_gQTL and each e_gQTL for the non-pancreatic tissues in GTEx. We identified 8 (31% of 16, all singletons) that were independent and exclusive to fetal-like iPSC-PPC (i.e., did not have LD; Table S16). One of these 8 iPSC-PPC e_gQTLs (TPD52 e_gQTL) is described below in further detail. These results show that integrating fetal-like iPSC-PPC eQTLs can help resolve certain GWAS loci that cannot be resolved using only adult datasets.”
---------------	---

4. Was there a rationale for performing all of the analyses on the hg19 genome build? This is now a somewhat
outdated genome build.

The iPSCORE resource has had molecular datasets generated over the past eight years. Here, we performed our
analyses on hg19 to be consistent with all of our other datasets in the iPSCORE resource. This enables us and
other researchers to perform multi-tissue genomic analyses for characterizing developmental genetic variants. We
will update all iPSCORE datasets to the new build soon.

**References:**

- 1. Wallace C. A more accurate method for colocalisation analysis allowing for multiple causal variants. Cordell
HJ, ed. *PLoS Genet.* 2021;17(9):e1009440. doi:10.1371/journal.pgen.1009440
- 2. Liang D, Elwell AL, Aygün N, et al. Cell-type-specific effects of genetic variation on chromatin accessibility
during human neuronal differentiation. *Nat Neurosci.* 2021;24(7):941-953. doi:10.1038/s41593-021-00858-
w

- 3. Currin KW, Erdos MR, Narisu N, et al. Genetic effects on liver chromatin accessibility identify disease
regulatory variants. *The American Journal of Human Genetics*. 2021;108(7):1169-1189.
doi:10.1016/j.ajhg.2021.05.001
- 4. Viñuela A, Varshney A, van de Bunt M, et al. Genetic variant effects on gene expression in human pancreatic
islets and their implications for T2D. *Nat Commun*. 2020;11(1):4912. doi:10.1038/s41467-020-18581-8
- 5. The GTEx Consortium. The GTEx Consortium atlas of genetic regulatory effects across human tissues.
*Science*. 2020;369(6509):1318-1330. doi:10.1126/science.aaz1776
- 6. Shaun Purcell CC. PLINK 1.9.0.
- 7. The 1000 Genomes Project Consortium, Corresponding authors, Auton A, et al. A global reference for human
genetic variation. *Nature*. 2015;526(7571):68-74. doi:10.1038/nature15393
- 8. Zhu Y, Liu Q, Zhou Z, Ikeda Y. PDX1, Neurogenin-3, and MAFA: critical transcription regulators for beta
cell development and regeneration. *Stem Cell Res Ther*. 2017;8(1):240. doi:10.1186/s13287-017-0694-z
- 9. Aigha II, Abdelalim EM. NKX6.1 transcription factor: a crucial regulator of pancreatic β cell development,
identity, and proliferation. *Stem Cell Res Ther*. 2020;11(1):459. doi:10.1186/s13287-020-01977-0
- 10. Seymour PA. Sox9: A Master Regulator of the Pancreatic Program. *Rev Diabet Stud*. 2014;11(1):51-83.
doi:10.1900/RDS.2014.11.51
- 11. Seymour PA, Freude KK, Tran MN, et al. SOX9 is required for maintenance of the pancreatic progenitor cell
pool. *Proc Natl Acad Sci USA*. 2007;104(6):1865-1870. doi:10.1073/pnas.0609217104
- 12. Yevtodiyyenko A, Schmidt JV. *Dlk1* expression marks developing endothelium and sites of branching
morphogenesis in the mouse embryo and placenta. *Dev Dyn*. 2006;235(4):1115-1123.
doi:10.1002/dvdy.20705
- 13. Rezanian A, Bruin JE, Arora P, et al. Reversal of diabetes with insulin-producing cells derived in vitro from
human pluripotent stem cells. *Nat Biotechnol*. 2014;32(11):1121-1133. doi:10.1038/nbt.3033
- 14. Gonçalves CA, Larsen M, Jung S, et al. A 3D system to model human pancreas development and its reference
single-cell transcriptome atlas identify signaling pathways required for progenitor expansion. *Nat Commun*.
2021;12(1):3144. doi:10.1038/s41467-021-23295-6
- 15. Geusz RJ, Wang A, Chiou J, et al. Pancreatic progenitor epigenome maps prioritize type 2 diabetes risk genes
with roles in development. *eLife*. 2021;10:e59067. doi:10.7554/eLife.59067
- 16. Ramond C, Beydag-Tasöz BS, Azad A, et al. Understanding human fetal pancreas development using
subpopulation sorting, RNA sequencing and single-cell profiling. *Development*. Published online January 1,
2018:dev.165480. doi:10.1242/dev.165480
- 17. Schmidt JV, Matteson PG, Jones BK, Guan XJ, Tilghman SM. The *Dlk1* and *Gtl2* genes are linked and
reciprocally imprinted. *Genes Dev*. 2000;14(16):1997-2002. doi:10.1101/gad.14.16.1997
- 18. Van Hoof D, D'Amour KA, German MS. Derivation of insulin-producing cells from human embryonic stem
cells. *Stem Cell Research*. 2009;3(2-3):73-87. doi:10.1016/j.scr.2009.08.003
- 19. Oliver-Krasinski JM, Stoffers DA. On the origin of the β cell. *Genes Dev*. 2008;22(15):1998-2021.
doi:10.1101/gad.1670808

- 20. Giambartolomei C, Vukcevic D, Schadt EE, et al. Bayesian Test for Colocalisation between Pairs of Genetic
Association Studies Using Summary Statistics. Williams SM, ed. *PLoS Genet.* 2014;10(5):e1004383.
doi:10.1371/journal.pgen.1004383
- 21. Yan J, Qiu Y, Ribeiro dos Santos AM, et al. Systematic analysis of binding of transcription factors to
noncoding variants. *Nature.* 2021;591(7848):147-151. doi:10.1038/s41586-021-03211-0
- 22. Thurner M, van de Bunt M, Torres JM, et al. Integration of human pancreatic islet genomic data refines
regulatory mechanisms at Type 2 Diabetes susceptibility loci. *eLife.* 2018;7:e31977. doi:10.7554/eLife.31977
- 23. Newman AM, Steen CB, Liu CL, et al. Determining cell type abundance and expression from bulk tissues
with digital cytometry. *Nat Biotechnol.* 2019;37(7):773-782. doi:10.1038/s41587-019-0114-2
- 24. Bioconductor Package Maintainer. liftOver: Changing genomic coordinate systems with rtracklayer::liftOver.
Published online 2022.

REVIEWERS' COMMENTS

Reviewer #1 (Remarks to the Author):

The authors have addressed all my comments. Congratulations on the work.

Reviewer #2 (Remarks to the Author):

The revisions made to this manuscript address the issues raised in the initial critique. The revisions have led to a more concise and clear presentation. The issues with confounding effects of different segmentation approaches were addressed well.

Reviewer #3 (Remarks to the Author):

In the revised manuscript by Nguyen et al., the authors have done a thorough job in addressing the reviewer comments, resulting in a much-improved and clearer manuscript. I have no further comments to the authors and would be happy to see this piece of work published.